# Distinct regulation of ATM signaling by DNA single-strand breaks and APE1

Haichao Zhao [1], Jia Li [1], Zhongsheng You [2], Howard D. Lindsay [3] & Shan Yan [1,4,5] ✉

In response to DNA double-strand breaks or oxidative stress, ATM-dependent DNA damage response (DDR) is activated to maintain genome integrity. However, it remains elusive whether and how DNA single-strand breaks (SSBs) activate ATM. Here, we provide direct evidence in *Xenopus* egg extracts that ATM-mediated DDR is activated by a defined SSB structure. Our mechanistic studies reveal that APE1 promotes the SSB-induced ATM DDR through APE1 exonuclease activity and ATM recruitment to SSB sites. APE1 protein can form oligomers to activate the ATM DDR in *Xenopus* egg extracts in the absence of DNA and can directly stimulate ATM kinase activity in vitro. Our findings reveal distinct mechanisms of the ATM-dependent DDR activation by SSBs in eukaryotic systems and identify APE1 as a direct activator of ATM kinase.

Cells are constantly exposed to various damaging insults such as ultraviolet light, chemical agents, reactive oxygen species (ROS), and internal metabolic dysfunctions, generating hundreds and thousands of DNA lesions per day[1–3]. To maintain genomic integrity, organisms activate evolutionarily conserved DNA damage response (DDR) pathways in response to DNA damage and/or stress conditions[4,5]. The DDR pathways generally involve DNA lesion recognition and a cellular signaling cascade to promote transcription activation, DNA repair, and cell cycle arrests[4,5]. The Ataxia-telangiectasia-mutated (ATM) and ataxia-telangiectasia and Rad3-related (ATR) proteins are major sensor kinases of DDR pathways, and play essential roles in maintaining genome integrity in eukaryotic cells[3,4,6]. ATM and ATR genes are expressed in most tissues, and mutations in ATM/ATR result in autosomal recessive disorders ataxia-telangiectasia[7] and Seckel syndrome[8], respectively. The ATM- or ATR-dependent DDR pathways primarily respond to DNA double-strand breaks (DSBs), or stalled DNA replication forks and DNA single-strand breaks (SSBs), respectively, whereas ATM and ATR often crosstalk with each under certain types of stress conditions and/or DNA lesions[3,9,10]. Activated ATM and ATR kinases phosphorylate their respective downstream substrates such as Chk1 and Chk2 to regulate cell cycle progression and promote DNA repair[4,5,11]. Both ATM and ATR DDR pathways are being targeted in clinical trials for diseases such as cancer, which highlights the significance of basic mechanistic studies on how ATM and ATR are activated to maintain genome stability[12].

Previous studies have elucidated several molecular mechanisms of how the ATM-dependent DDR is activated. It is well-documented that ATM is activated by DSBs[13–16]. The Mre11-Rad50-Nbs1 (MRN) complex recognizes DSB ends and promotes the recruitment of ATM to DSBs for ATM activation and DSB repair[16–20]. ATM phosphorylation at Ser1981 is rapidly induced by DSBs in trans, and has been widely utilized as an indicator of ATM activation[13,17,21]. Whereas DSBs and the MRN complex directly active ATM kinase in vitro[14,16], Mre11 and Nbs1 are also phosphorylated by activated ATM kinase to participate in the downstream signaling for DNA repair and cell cycle checkpoints[22–24]. Furthermore, histone acetyltransferase Tip60 and its cofactor TRRAP promote ATM activation at DSB sites via ATM interaction and acetylation at Lys3016[25–27]. In addition, oxidative stressor hydrogen peroxide can directly activate ATM through distinct ATM homodimerization Cys2991-mediated disulfide bond formation independent of DNA or the MRN complex[28–30]. Of note, the ATM kinase can be directly activated by the purified MRN protein complex in in vitro reconstitution system without DNA[14]. Overall, ATM signaling can be activated by DSBs, oxidative stress, or its direct activator proteins in various experimental systems.

SSBs can arise from oxidized nucleotides or bases, intermediate products of DNA repair, or abortive cellular enzymes such as DNA

[1]Department of Biological Sciences, University of North Carolina at Charlotte, Charlotte, NC 28223, USA. [2]Department of Cell Biology and Physiology, Washington University School of Medicine, St. Louis, MO 63110, USA. [3]Lancaster Medical School, Faculty of Health and Medicine, Lancaster University, Lancaster LA1 4YQ, UK. [4]School of Data Science, University of North Carolina at Charlotte, Charlotte, NC 28223, USA. [5]Center for Biomedical Engineering and Science, University of North Carolina at Charlotte, Charlotte, USA. ✉e-mail: shan.yan@charlotte.edu

topoisomerase 1, and can be generated more than 10,000 times per day each cell[3,31,32]. Accumulation of unrepaired SSBs has been implicated in the pathogenesis of human diseases such as cancer and heart failure[3,31–35]. To maintain genome stability, the ATR DDR signaling is activated by SSBs derived from damaged chromatin DNA under stress conditions (e.g., methyl methanesulfonate (MMS) and hydrogen peroxide) or defined SSB structures in various experimental systems[31,34–39]. Accumulating evidence suggests that ATM-mediated DDR signaling can also be activated by SSBs derived from DNA oxidative/alkylation lesions or presumptive SSBs arising in XRCC1-knockdown mammalian cells[33,40–42]. Furthermore, targeted induction of 8-oxoguanine (8-oxoG) at telomeres in human fibroblast and epithelial cells can trigger activation of the ATM-Chk2 DDR pathway and p53-dependent senescence[43]. However, it remains unclear whether ATM DDR signaling is activated directly by defined SSB structures, or indirectly from DSBs converted from SSBs when meeting with ongoing DNA replication forks.

Various *Xenopus laevis* egg extract systems including Low-Speed Supernatant (LSS), High-Speed Supernatant (HSS), and Nucleo-Plasmic Extract (NPE) have been developed and widely utilized in structure and function studies of DNA repair (e.g., DNA inter-strand crosslink (ICL) repair and DSB repair) and DDR pathways (e.g., ATR and ATM)[20,44–48]. Briefly, eggs from female African clawed frogs are crushed after a series of centrifugations at low speed (20,000 g) and high speed (260,000 g) to make the HSS, which contains most (if not all) proteins derived from eggs (e.g., replication/repair/DDR proteins) but not endogenous chromatin DNA, organelles (e.g., mitochondria and ribosomes), yolk, pigment granules, and membrane fractions[45,49–51]. Purified plasmid DNA containing no apparent lesions (wild type, WT) or defined site-specific DNA damage (e.g., ICL, DSB, SSB) can be added to the HSS as exogenous DNA source for replication/repair/DDR analyses; however, plasmid DNA in the HSS system can potentially form pre-Replication Complexes but can't continue DNA replication elongation until the addition of NPE which provides the replication elongation-needed S-CDK and DDK kinase activities[49,50,52]. Several advantages of using the HSS system include loss of function characterization (e.g., immuno-depletion of target protein of interest or adding small molecule inhibitor) and gain of function or rescue experiment analyses (e.g., adding purified WT or mutant recombinant proteins at desired concentrations)[45,49,50,52].

Apurinic/apyrimidinic endonuclease 1 (APE1, also known as APEX1 or APN1) displays AP endonuclease, 3′–5′ exonuclease, 3′-phosphodiesterase, and 3′ exoribonuclease activities, and plays a central role in the removal and repair of numerous oxidative DNA lesions including AP sites[53–57]. In addition to the well-characterized roles in base excision repair (BER), redox regulation, and immune response[55,58–60], accumulating evidence supports APE1's previously uncharacterized but critical roles in DDR signaling pathways. ATR-mediated DDR pathway is activated by SSBs derived from oxidative damage on chromatin DNA in *Xenopus* LSS system and defined plasmid-based SSB structures in *Xenopus* HSS system[36,37,61–63]. Our recent studies using *Xenopus* HSS system have revealed that APE1 promotes the activation of ATR-Chk1 DDR pathway induced by defined SSB structure and defined ssDNA gap structures[63,64]. APE1 also promotes SSB repair by initiating the 3′–5′ SSB end resection process through its distinct 3′–5′ exonuclease activity in *Xenopus* HSS system[63]. In addition, APE1 is important for the activation of ATR-Chk1 DDR pathway under stress conditions (e.g., hydrogen peroxide and MMS) in human cancer cells, suggesting a conserved role of APE1 in the ATR-dependent DDR during evolution[65]. Intriguingly, overexpressed APE1 assembles into biomolecular condensates, which promotes the ATR-Chk1-dependent DDR under unperturbed conditions in the nucleolus of cancer cells in an APE1 nuclease activity-independent manner[65]. APE1 was also shown to modulate the ATM-Chk2 DDR pathway activated by DNA alkylation damage in cultured human cancer HeLa cells, likely due to the

generation of SSBs from AP sites by APE1's AP endonuclease activity[42]. Nonetheless, it remains unknown whether and how APE1 regulates ATM activation induced by SSBs or other activator proteins other than MRN complex to maintain genome integrity.

In this study, we provide direct evidence that the ATM-dependent DDR pathway is activated by a SSB with a defined structure in *Xenopus* egg extracts. Our results indicate that SSB induces ATM activation prior to ATR activation, and that SSB-induced ATM activation is important for the dynamic regulation of SSB repair. We demonstrate that APE1 promotes the SSB-induced ATM DDR through at least two mechanisms: APE1 exonuclease activity-mediated SSB processing and APE1-mediated direct recruitment of ATM to SSBs. Notably, we show that addition of purified recombinant APE1 protein induces the activation of ATM DDR pathway in the absence of DNA in *Xenopus* egg extracts and that recombinant APE1 protein can directly stimulate ATM kinase to phosphorylate its substrate Chk2 in in vitro kinase assays, suggesting APE1 can act as a direct activator of ATM kinase. Moreover, the N-terminal motif (NT34) and especially the positively charge lysine residues of APE1 are critical for APE1 dimerization/oligomerization and ATM activation. Taken together, our results have uncovered mechanisms of ATM activation by SSBs and identified APE1 as a direct ATM activator.

## Results
### ATM DDR pathway is activated by a defined SSB in the *Xenopus* HSS system

Previous studies have shown that plasmid-based DSB structures trigger ATM DDR activation, as indicated by phosphorylation of *Xenopus* ATM at Ser1989 (designated as P-ATM, homologous to human ATM at Ser1981) and phosphorylation of *Xenopus* H2AX at Ser136 (designated as γH2AX, homologous to human H2AX at Ser139), in *Xenopus* LSS and HSS systems[17,21,66,67]. To investigate SSB repair and DDR signaling in *Xenopus* HSS system, we generated plasmid-based defined SSB structure containing 3′- and 5′-hydroxyl (OH) group and DSB structures at defined locations, as recently described[36,37,63]. To determine whether a defined SSB structure triggers ATM activation, we incubated different concentrations of SSB or Control (CTL) plasmid in the HSS for 30 min. We found that the SSB plasmid, but not CTL plasmid, at the concentrations of 20 ng/μL (~12 nM) and 40 ng/μL triggered robust ATM phosphorylation and γH2AX in the HSS system (Fig. 1A). It was noticed that ATM phosphorylation and γH2AX were also elevated to some extent by CTL plasmid at a higher concentration (i.e., 80 ng/μL) (Fig. 1A), likely due to contaminating SSB in the preparation of CTL plasmid (Fig. S1A). The quality and purity of SSB structure and DSB structure were confirmed by agarose gel electrophoresis and ethidium bromide staining (Fig. S1A). Our time-course experiment showed that SSB-induced ATM phosphorylation appeared at ~5 min and continued to increase during the incubation in the HSS and that γH2AX occurred with even faster kinetics, compared with ATM phosphorylation (Fig. 1B). These observations suggest that the defined SSB structure induces the activation of the ATM DDR pathway in a dose- and time-dependent manner in the *Xenopus* HSS system.

Consistent with prior studies[17,20], DSB-containing plasmid but not CTL plasmid structure induced ATM phosphorylation and γH2AX in a dose- and time-dependent manner in a *Xenopus* HSS system (Fig. S1B, C). We found that a final concentration of 10 ng/μL of DSB-containing plasmid in HSS was enough to induce ATM phosphorylation efficiently, whereas 5 ng/μL DSB-containing plasmid was sufficient for robust γH2AX activation (Fig. S1B). ATM phosphorylation and γH2AX appeared at as early as 5 min after 10 ng/μL DSB plasmid was incubated in the HSS (Fig. S1C). Moreover, potential DSB contamination in our SSB plasmid preparation is typically less than ~1% (Fig. S1A). These control experiments exclude the possibility of confounding DSB in our SSB preparation for ATM DDR activation induced by the SSB-containing plasmid. Due to the absence of DNA replication in

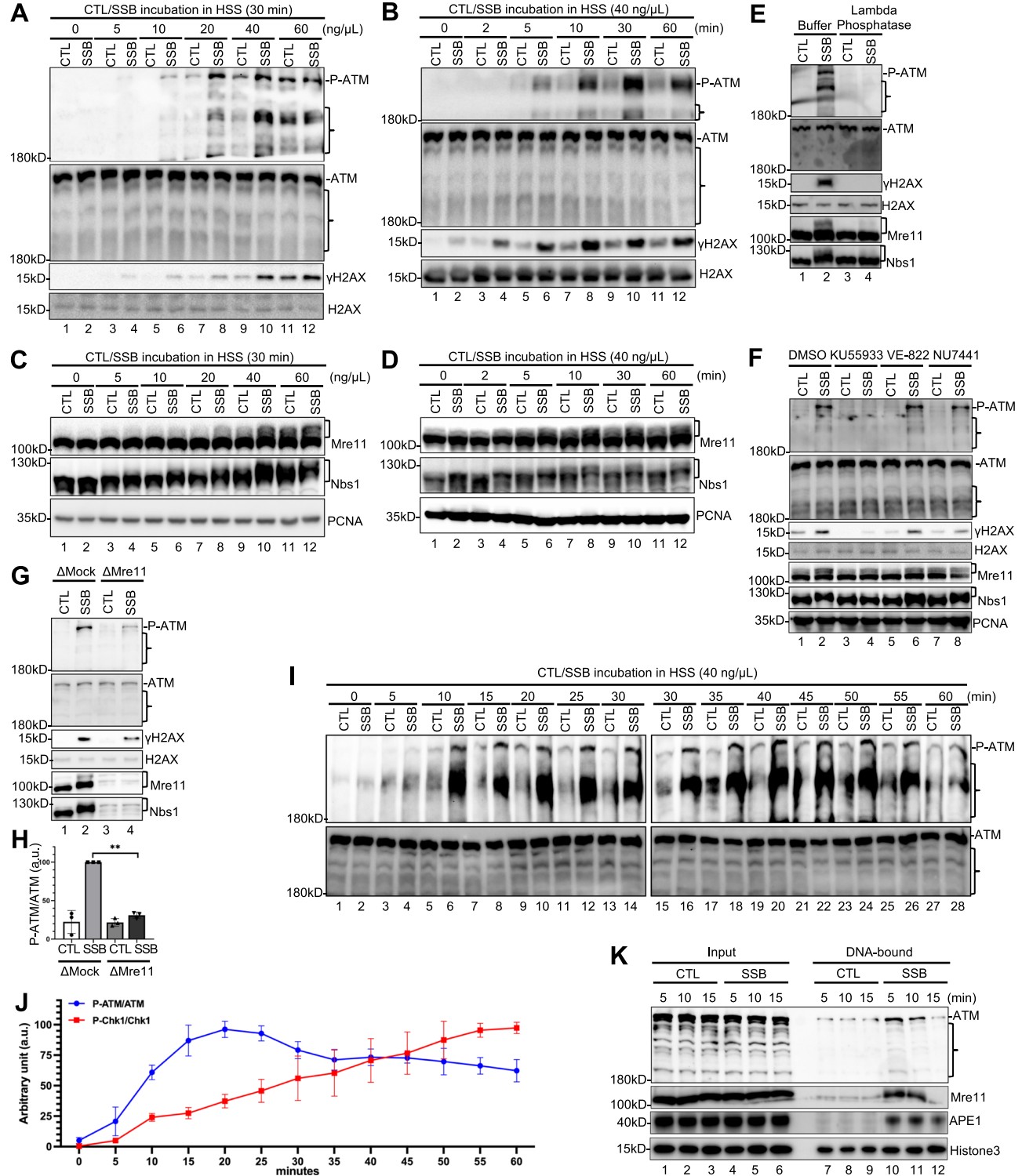

*Xenopus* HSS[37,49], our SSB-induced ATM DDR system also excludes the possibility of SSB-to-DSB conversion due to DNA replication.

Previous studies have shown that DSBs induce Mre11 and Nbs1 phosphorylation mediated by ATM in *Xenopus* egg extracts[68,69]. We found that SSB plasmid, but not CTL plasmid, induced mobility shifts of Mre11 and Nbs1 in the HSS in a dose- and time-dependent manner (Fig. 1C, D). To confirm the mobility shifts of Mre11 and Nbs1 induced by SSB plasmid were due to phosphorylation, we pre-treated the HSS with Lambda Phosphatase and found that the SSB-induced Mre11/Nbs1 mobility shifts were largely compromised (Fig. 1E). This observation suggests that SSB but not CTL plasmid triggers Mre11 and Nbs1

phosphorylation in the HSS system (indicated by their mobility shifts on gel). As expected, the SSB-induced ATM phosphorylation and γH2AX were also impaired by Lambda Phosphatase (Fig. 1E). To further determine whether the SSB-induced phosphorylation of ATM, H2AX, Nbs1 and Mre11 is dependent on ATM kinase in the HSS, we pre-incubated KU55933 (ATM kinase specific inhibitor)[21,70], VE-822 (ATR kinase specific inhibitor)[37,71], or NU7441 (DNA-PK kinase specific inhibitor)[19,72,73] in the HSS prior to the addition of SSB or CTL plasmid (Fig. 1F). We found that KU55933 but neither VE-822 nor NU7441 compromised the SSB-induced phosphorylation of ATM, Nbs1, Mre11 and H2AX, suggesting that the defined SSB structure can induce ATM-

**Fig. 1 | ATM is activated by a defined SSB structure in the HSS system. A** CTL or SSB plasmid was incubated with HSS at indicated concentrations for 30 min at room temperature. Extracts were examined via immunoblotting. **B** CTL or SSB plasmid was added to HSS at a final concentration of 40 ng/μL. After different time of incubation at room temperature, the extracts were examined via immunoblotting. **C** CTL or SSB plasmid was added to HSS at different concentrations as indicated. Extracts were examined via immunoblotting analysis. **D** CTL or SSB plasmid was added to HSS at a final concentration of 40 ng/μL. After different time of incubation, the extracts were examined via immunoblotting. **E** Lambda phosphatase was added to HSS supplemented with CTL or SSB plasmid. After a 15-minute incubation, the extracts were examined via immunoblotting. **F** ATM inhibitor KU55933 (1 mM), ATR inhibitor VE-822 (1 mM), or DNA-PK inhibitor NU7441 (25 μM) was added to HSS supplemented with CTL or SSB plasmid. After incubation, the extracts were examined via immunoblotting. **G** CTL or SSB plasmid was added to Mock- or Mre11-depleted HSS. After incubation, the extracts were examined via immunoblotting. **H** Quantification and statistical analysis of P-ATM/ATM shown in Panel G. Data are presented as mean values ± SD. **$p$ = 0.0011 (two-tailed, paired t-test). n = 3. a.u., arbitrary units. **I** CTL or SSB plasmid was added to HSS at a final concentration of 40 ng/μL. After different times of incubation, the extracts were examined via immunoblotting. **J** Quantification of intensity of protein bands from Fig. 1I& Fig. S1I, and ratios of P-ATM/ATM, and P-Chk1/Chk1 were shown. These samples derive from the same experiment and blots were processed in parallel. Data are presented as mean values ± SD. n = 3. **K** CTL or SSB plasmid was added to HSS at 40 ng/μL. After different incubation times, the extracts and DNA-bound fractions were examined via immunoblotting. The data presented in Panel **A–G**, **I**, and **K** are representative of three biological replicates. ] in Panel **C–G** indicates mobility shifts of Mre11 and Nbs1. } indicates non-specific bands. Source data are provided as a Source Data file.

dependent DDR signaling in the HSS system (Fig. 1F). Additionally, VE-822 pretreatment compromised the SSB-induced Chk1 phosphorylation at Ser344 (designated as P-Chk1, homologous to human Chk1 at Ser345) and RPA32 phosphorylation at Ser33 (designated as P-RPA32) (Fig. S1D).

To test whether Mre11 affects the SSB-induced ATM DDR, we took two approaches: immuno-depletion of Mre11 from the HSS, and addition of Mirin, a specific inhibitor of Mre11's exonuclease activity[19,74]. We found that the SSB-induced ATM phosphorylation and γH2AX were compromised when Mre11 was immunodepleted from the HSS (Fig. 1G, H). The majority of Nbs1 was co-depleted from the HSS with anti-Mre11 antibodies, suggesting that the MRN complex is important for the activation of the SSB-induced ATM DDR (Fig. 1G, H). Consistently, the addition of Mirin significantly impaired SSB-induced phosphorylation of ATM, H2AX and Mre11 in the HSS system, suggesting that the exonuclease activity of Mre11 is important for the SSB-induced ATM DDR (Fig. S1E, F). Additionally, we observed similar results when Nbs1 was immunodepleted from the HSS with anti-Nbs1 antibodies (Fig. S1G). Together, our results indicate that ATM DDR pathway activated by a defined SSB structure requires the MRN complex in the *Xenopus* HSS system.

### ATM DDR activation by SSBs precedes ATR DDR activation in the HSS system

We have recently shown that the ATR DDR pathway is activated by a defined SSB in the HSS system[37,63]. This prompted us to examine the relationship between ATM activation and ATR activation in response to SSBs. To directly compare the kinetics of the activation of the SSB-induced ATM and ATR DDR in the HSS system, we added SSB or CTL plasmid at a final concentration of 40 ng/μL and collected samples for immunoblotting analysis every 5 min during incubation (Fig. 1I and S1I). Our quantitative results showed that the SSB-induced ATM phosphorylation rapidly peaked at ~20 min, and decreased slowly thereafter (Fig. 1J). The SSB-induced Chk1 phosphorylation continuously accumulated to peak at ~55-60 min (Fig. 1J and S1I). These observations suggest that ATM DDR activation may precede that of ATR DDR. Consistent to this sequential ATM-ATR activation by SSB, inhibition of ATR by VE-822 was almost dispensable for the SSB-induced ATM phosphorylation and γH2AX (Fig. 1F), whereas inhibition of ATM by KU55933 partially decreased the SSB-induced Chk1 and RPA32 phosphorylation (Fig. S1D).

We have established a DNA-bound fraction isolation approach in *Xenopus* system and reported that APE1 was recruited to SSB sites at early timepoints of incubation (i.e., 5, 10, and 15 min) in the HSS system (also shown in Fig. 1K)[63]. Importantly, ATM and Mre11 were recruited onto SSB plasmid but not CTL plasmid at 5 and 10 min after incubation in the HSS system (Fig. 1K), suggesting that ATM and the MRN complex are recruited to SSB sites for ATM activation at an early stage of SSB signaling. Our control experiment showed that almost no ATM or APE1 was detected in DNA-bound fractions when no DNA was added in the HSS system (Fig. S1H). Intriguingly, DNA-bound ATM and Mre11 at SSB sites were decreased at 15-min timepoint, while phosphorylated ATM was observed after 10-min incubation in total egg extracts (Fig. 1I–K). We speculate that after phosphorylation and activation at SSB sites, ATM and Mre11 may disengage from the DNA into extracts and retain their phosphorylated and activated status. Our observations here reminded us of a previous study showing that ATM and Mre11 are recruited to DSB sites for activation and phosphorylated ATM and Mre11 dissociate from DNA in *Xenopus* egg extracts[22].

### The SSB-induced ATM DDR regulates SSB repair efficiency in the HSS system

To elucidate the significance of SSB-induced ATM activation, we tested whether the efficiency and speed of SSB repair is affected by ATM DDR in the HSS system. Similar to a previous observation[37], most of the SSB structure (~80%) can be repaired from nicked version into circular version in 30 min in a VE-822-sensative manner in the HSS system (Fig. S2A, B). The addition of ATM inhibitor KU55933 significantly decreased SSB repair efficiency (Fig. 2A, B). As previously demonstrated[21], endogenous ATM can be depleted from HSS with anti-ATM antibodies (Fig. S2C). ATM depletion also significantly decreased the repair of the SSB structure in the HSS system (Fig. 2C, D). These observations suggest that ATM and its kinase activity promote SSB repair in the HSS system.

### APE1, but not APE2, is critical for SSB-induced ATM DDR signaling in the HSS system

To elucidate the mechanism of ATM activation by SSBs, we tested the involvement of APE1 and APE2. APE1 and APE2 resect SSBs in the 3′–5′ direction sequentially via their exonuclease activity to promote the ATR DDR pathway in the HSS system[37,63]. Interestingly, the SSB-induced ATM phosphorylation was abrogated in APE1-depleted HSS (Fig. 2E). In contrast, no obvious changes for the SSB-induced ATM phosphorylation were observed in APE2-depleted HSS compared with mock-depleted HSS (Fig. S2D). Pretreatment of the HSS with AR03, an APE1 inhibitor that inhibits both AP endonuclease and exonuclease activities of APE1[63,75], also significantly impaired SSB-induced ATM and Nbs1 phosphorylation and γH2AX (Fig. S2E). Notably, APE1 depletion had only mildly decreased to almost no noticeable effect on the DSB-induced ATM phosphorylation and γH2AX (Fig. S2F). These observations strongly suggest that APE1, but not APE2, is critical for SSB-induced ATM DDR signaling and that APE1 is required for ATM DDR activation by SSBs but not by DSBs.

We have showed that APE1 is recruited to SSB plasmid at an early stage of SSB signaling (Fig. 2F)[63]. Thus, we asked whether APE1 affects the recruitment of ATM and Mre11 to SSB sites. Indeed, the recruitment of ATM and Mre11 to SSB sites significantly decreased in APE1-depleted HSS, compared to Mock-depleted HSS (Fig. 2F). Similarly, pretreatment of the HSS with APE1 inhibitor AR03 also decreased the recruitment of ATM and Mre11 to SSB sites in a dose-dependent manner

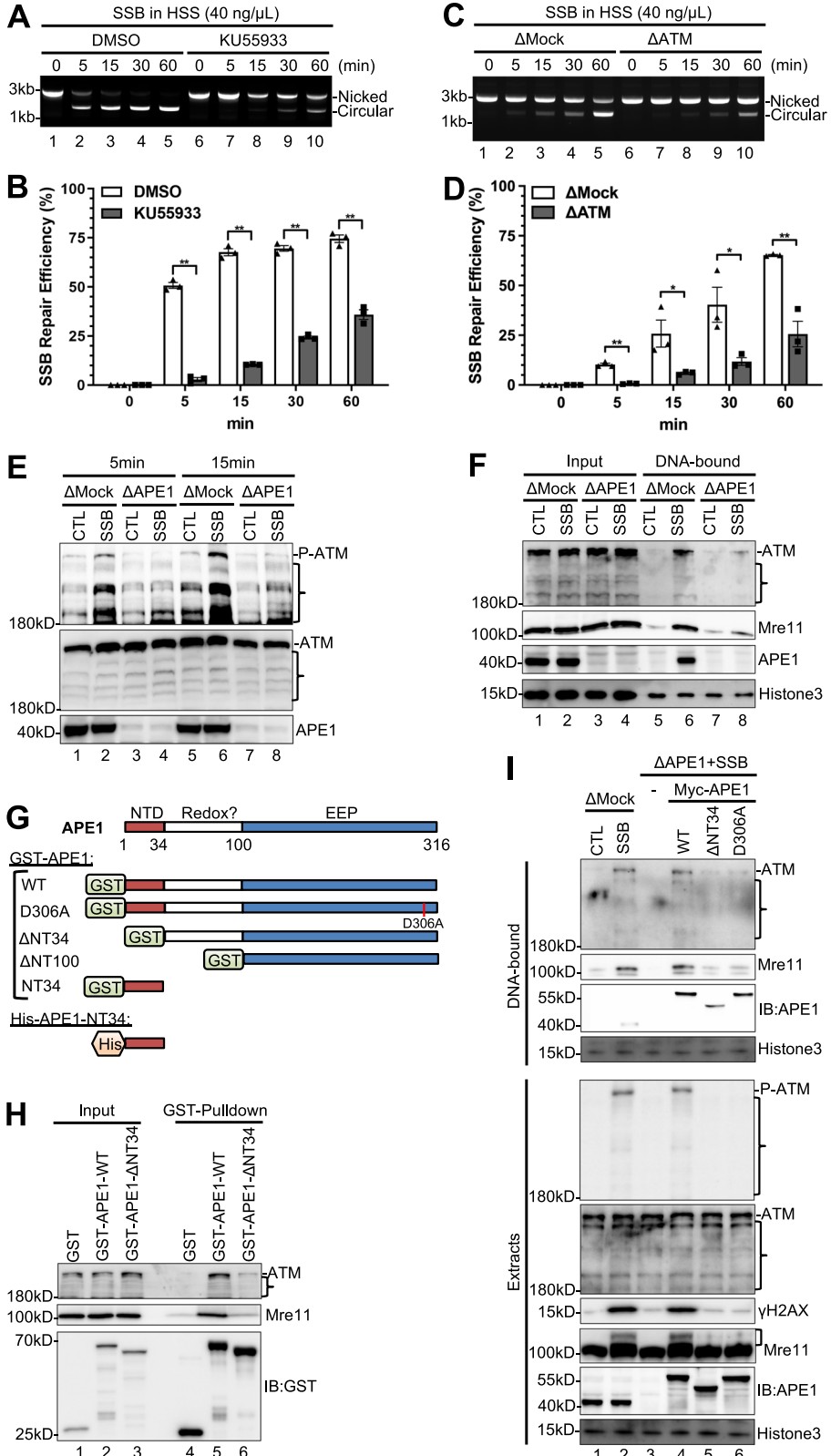

(Fig. S2H). These observations strongly suggest that APE1 is crucial for the recruitment of ATM and MRN complex to the defined SSB structure in the HSS system. In contrast, ATM inhibitor KU55933 had almost no noticeable effect on the recruitment of APE1 to SSB sites (Fig. S2G), suggesting that APE1 association onto SSB sites is not regulated by ATM kinase as a potential feedback mechanism after its activation.

We have recently shown that the NT33 motif within APE1 is required for its interactions with ATR, ATRIP, and RPA to regulate the ATR-Chk1 DDR pathway in cells and nuclear extracts[65,76]. To further elucidate the underlying mechanism of APE1's function in SSB-induced ATM DDR, we sought to test whether APE1 interacts with ATM to facilitate its recruitment onto SSB sites. Our GST-pulldown experiments show that GST-APE1-WT, but not GST nor GST-APE1-ΔNT34 lacking the N-terminal 34 amino acids (NT34), could pulldown endogenous ATM and Mre11 in the HSS (Fig. 2G, H and S2I). This domain mapping experiment reveals the significance of NT34 domain of APE1

**Fig. 2 | APE1 is critical for ATM activation and SSB repair in the HSS system.**
**A** The SSB plasmid was added to HSS supplemented with the ATM inhibitor KU55933 at a final concentration of 1 mM. After different incubation times, the DNA repair products were isolated and analyzed on an agarose gel. **B** Quantification results from Fig. 2A. Data are presented as mean values ± SD. **p(5 min)=0.0007; **p(15 min)=0.0011; **p(30 min)=0.0022; **p(60 min)=0.0087; two-tailed, paired t-test, n = 3. **C** The SSB plasmid was added to Mock- or ATM-depleted HSS. After different incubation times, the DNA repair products were isolated and analyzed on an agarose gel. **D** Quantification results from Fig. 2C. Data are presented as mean values ± SD. **p(5 min)=0.0002; *p(15 min)=0.0443, *p(30 min)=0.0337; **p(60 min)=0.0035; two-tailed, unpaired t-test, n = 3. **E** CTL or SSB plasmid was added to Mock- or APE1-depleted HSS. After different incubation times (5 and 15 min), the egg extracts were examined via immunoblotting analysis as indicated. **F** CTL or SSB

plasmid was added to Mock- or APE1-depleted HSS at 40 ng/μL. After a 10-min incubation, the total egg extracts ("Input") and DNA-bound fractions were examined via immunoblotting. **G** Schematic diagram of mutants and truncations of GST-APE1 proteins and His-APE1-NT34. **H** GST, WT or ΔNT34 GST-APE1 was added to HSS. After 30-min incubation, GST beads were added into the mixture and incubated at 4 °C. The total egg extracts ("Input") and GST-Pulldown fractions were examined via immunoblotting. **I** CTL or SSB plasmid was added to Mock-depleted HSS. SSB plasmid was added to APE1-depleted HSS supplemented with negative control (−), WT, ΔNT34, and D306A Myc-APE1 protein. After incubation, the total egg extracts ("Extracts") and DNA-bound fractions were examined via immunoblotting. ] indicates mobility shift of Mre11. The data presented in Panel **A, C, E, F, H, I** are representative of three biological replicates. } indicates non-specific bands. Source data are provided as a Source Data file.

in the interaction with ATM and Mre11 (Fig. 2H). Notably, our rescue experiment shows that adding back WT Myc-APE1, but not ΔNT34 Myc-APE1, to APE1-depleted HSS rescued the SSB-induced ATM phosphorylation and γH2AX (Fig. 2I). ΔNT34 Myc-APE1 also failed to rescue the recruitment of ATM and Mre11 onto SSB sites in APE1-depleted HSS (Fig. 2I). These observations suggest that APE1 may interact with ATM and promote its recruitment onto SSB sites for activation.

We next further test the role of APE1 exonuclease activity in the SSB-induced ATM DDR. To this end, we took advantage of a previously characterized mutant D306A APE1 which is exonuclease-deficient but AP endonuclease-proficient (Fig. 2G)[63]. We found that WT Myc-APE1 but not D306A Myc-APE1 rescued the SSB-induced ATM phosphorylation and γH2AX in APE1-depleted HSS (Fig. 2I). Although both WT and D306A Myc-APE1 associated with SSB sites similarly, WT Myc-APE1 but not D306A Myc-APE1 rescued the recruitment of ATM and Mre11 onto SSB sites in APE1-depleted HSS (Fig. 2I). These observations underline the significance of APE1 exonuclease activity for its function in SSB-induced ATM DDR pathway.

## Excess APE1 protein can activate ATM DDR signaling in the absence of DNA in the HSS system

Human APE1 forms biomolecular condensates and interacts with ATR and ATRIP to promote the ATR DDR signaling pathway[65]. Because APE1 also interacted with ATM in the HSS (Fig. 2H), we hypothesized that APE1 might also undergo oligomerization when ectopically over-expressed to activate ATM DDR signaling. To test this idea, we incubated different concentrations of GST-APE1 or GST protein in the HSS for 30 min and tested ATM DDR activation (Fig. 3A). Our results indicated that 16 μM of GST-APE1, but not GST, triggered robust ATM and Mre11 phosphorylation and γH2AX, indicative of ATM DDR activation (Fig. 3A). γH2AX was activated by GST-APE1 even at lower concentrations (2 and 4 μM) (Fig. 3A). Our time-course experiment showed that GST-APE1 readily activated ATM at ~5-10 min, and phosphorylation of Nbs1 and Mre11 and γH2AX at ~5 min (Fig. 3B). To further determine whether GST-APE1 induces a unique ATM-dependent DDR signaling pathway in the HSS, we pre-incubated KU55933, VE-822, or NU7441 in the HSS before the excess addition of GST or GST-APE1 proteins and revealed that the addition of KU55933, but not VE-822 or NU7441, compromised the GST-APE1-induced phosphorylation of ATM, Nbs1 and H2AX, suggesting that APE1 can directly activate the ATM-dependent DDR signaling pathway in the HSS system (Fig. 3C). To rule out any potential compounding effects of DNA and RNA in the HSS system, we treated the HSS with DNase I or RNase A separately or in combination, and found that GST-APE1 still induced phosphorylation of ATM and Nbs1 as well as γH2AX in the presence of the nucleases (Fig. 3D). These data suggest that excess APE1 protein can directly activate ATM-mediated DDR signaling in the HSS system.

It has been shown that purified His-tagged recombinant human APE1 protein can activate the ATR-Chk1 DDR pathway in nuclear extract of cultured human cancer cells[65]. We wanted to test whether ectopically overexpressed GST-APE1 protein in *Xenopus* HSS can also

trigger ATR DDR signaling activation. Similar to the finding in the human system[65], addition of GST-APE1, but not GST, to the HSS triggered Chk1 phosphorylation and RPA32 phosphorylation in a dose- and time-dependent fashion, suggesting that recombinant APE1 protein can also activate ATR-Chk1 DDR signaling pathway in the *Xenopus* HSS system (Fig. S3A, B). ATR inhibitor VE-822 impaired GST-APE1-induced phosphorylation of Chk1 and RPA32, but not phosphorylation of ATM, Nbs1, and H2AX (Fig. 3C). In contrast, KU55933 inhibited all phosphorylation events of these ATM/ATR indicator proteins (i.e., ATM, Nbs1, H2AX, Chk1 and RPA32) induced by GST-APE1 in the HSS (Fig. 3C). These observations suggest that ectopically overexpressed APE1-induced ATM signaling may play an upstream role of ATR signaling in the HSS system when DNA is absent.

Our domain mapping experiments revealed that excess addition of ΔNT34 and ΔNT100 GST-APE1 failed to induce phosphorylation of ATM, Mre11, Nb11, Chk1, RPA32 and H2AX, suggesting that the NT34 motif of APE1 is required for its function in the activation of ATM and ATR DDR (Figs. 2G, 3E, and S3C, D). Compared to WT GST-APE1 (16 μM), excess addition of NT34 GST-APE1 protein at very high concentrations (100 and 200 μM) also resulted in phosphorylation of ATM, Chk1 and H2AX, suggesting that the NT34 motif of APE1 is sufficient for ATM DDR activation in the HSS when DNA is absent (Fig. 3E, F). We previously reported that the W119R mutation within human APE1 impaired its ability to directly activate ATR DDR in cancer cells[65]. Interestingly, GST-APE1-W118R (homologous to human APE1 W119R) was defective for Chk1 phosphorylation but proficient in inducing ATM phosphorylation and γH2AX in the HSS (Fig. S3E). These data suggest that APE1 may promote the activation of ATM and ATR DDR signaling pathways via distinct mechanisms. Additionally, overexpression of D306A GST-APE1 in the HSS activated the phosphorylation of ATM and H2AX and enhanced the mobility shifts of Mre11 and Nbs1, in a similar fashion as WT GST-APE1 control, suggesting that APE1's exonuclease is dispensable for APE1-induced ATM signaling when DNA is absent in the HSS (Fig. S3F). Overall, observations here and findings from our previous studies[65] have revealed that APE1 NT34 motif is required for the activation of ATM and ATR DDR signaling and that APE1's AAD domain containing W119 is important for ATR signaling but dispensable for ATM signaling in the absence of DNA in the HSS system.

## APE1 directly stimulates ATM kinase activity in vitro

To determine whether APE1 can directly activate ATM kinase in vitro, we aimed to set up ATM kinase activity assays as previously described with some modifications[14,16,77]. Chk2 is an ATM substrate, and Chk2 phosphorylation at T68 is widely accepted as an indicator of ATM kinase activation[11,78]. Thus, we purified His-tagged human Chk2 (His-hChk2, kinase dead D347A) protein from bacteria (Fig. S4A), and incubated with Flag-tagged human ATM (Flag-hATM) purified from human cells (Fig. S4B). Our in vitro kinase assays showed that phosphorylation of His-hChk2 protein by Flag-hATM was gradually enhanced by increasing concentrations of His-hAPE1 (Fig. 4A). Furthermore, such His-hAPE1-stimulated phosphorylation of His-hChk2

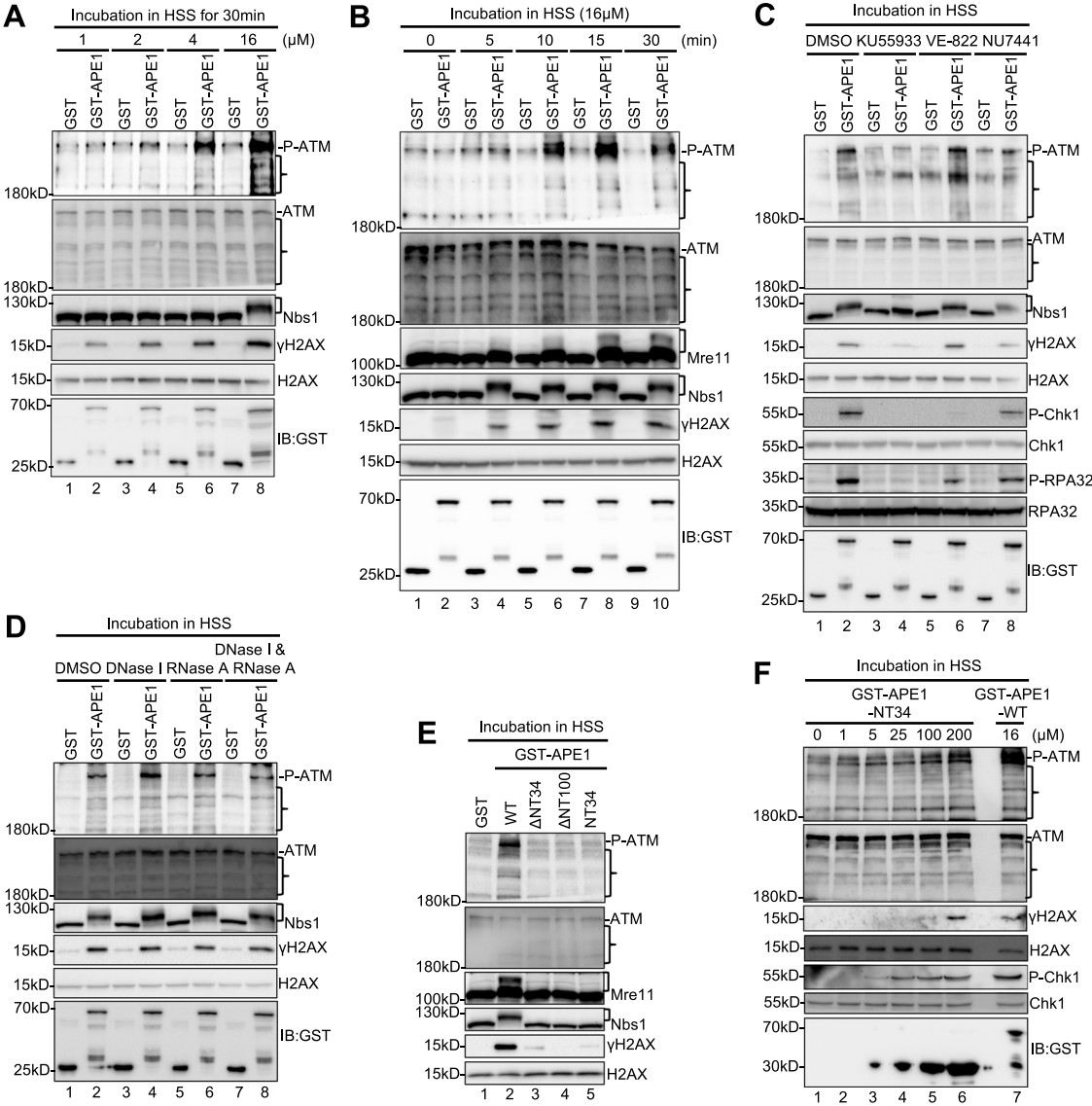

**Fig. 3 | APE1 actives the ATM DDR pathway in the absence of DNA in the HSS system. A** GST or GST-APE1 was added to HSS at different concentrations as indicated, and incubated for 30 min. Extracts were examined via immunoblotting analysis for ATM pathway as indicated. **B** GST or GST-APE1 was added to HSS at a final concentration of 16 μM. After different times of incubation at room temperature, the extracts were examined via immunoblotting analysis as indicated. **C** ATM inhibitor KU55933 (1 mM), ATR inhibitor VE-822 (1 mM), or DNA-PK inhibitor NU7441 (25 μM) were added to HSS supplemented with GST or GST-APE1 (16 μM) and incubated for 30 min. Extracts were examined via immunoblotting analysis as indicated. **D** DNase I, RNase A, or DNase I & RNase A was added to HSS supplemented with GST or GST-APE1 (16 μM) for 30 min. Extracts were examined via immunoblotting analysis as indicated. **E** GST or GST-APE1-WT, ΔNT34, ΔNT100, or NT34 was added to HSS at a final concentration of 16 μM, and incubated for 30 min. Extracts were examined via immunoblotting analysis for ATM pathway as indicated. **F** WT or NT34 GST-APE1 was added to HSS at different concentrations as indicated, and incubated for 30 min. Extracts were examined via immunoblotting analysis for ATM/ATR pathway as indicated. ] in Panel **A**–**E** indicates mobility shift of Mre11 or Nbs1. The data presented are representative of three biological replicates. } indicates non-specific bands. Source data are provided as a Source Data file.

was compromised by the addition of ATM inhibitor KU55933 (Fig. 4A). Flag-hATM-KD (kinase dead mutant, D2870A/N2875K) was used as a negative control for the in vitro kinase assay (Fig. 4B). These observations indicate that purified recombinant hAPE1 protein can directly stimulate ATM kinase activity in in vitro kinase assays.

We next tested whether *Xenopus* APE1 played a conserved function in ATM kinase activation using the established in vitro kinase assays. Our results showed that WT GST-APE1, but not GST, also stimulated His-hChk2 phosphorylation by Flag-hATM (Fig. 4C, D), and that such *Xenopus* APE1-stimulated Chk2 phosphorylation was abolished by the ATM inhibitor KU55933 or when Flag-hATM-KD was utilized (Fig. 4C, D). ΔNT34 GST-APE1 was much less efficient in stimulating the phosphorylation of His-hChk2 by Flag-hATM, suggesting that the NT34 motif of *Xenopus* APE1 is required for the direct

activation of ATM kinase by APE1 (Fig. 4C–E). We also purified His-tagged *Xenopus* APE1 NT34 (His-APE1-NT34) (Fig. S4C) and found that His-APE1-NT34 protein at high concentrations (100 and 300 μM) also stimulated phosphorylation of His-hChk2 by Flag-hATM in an KU55933-sensative manner, suggesting that the NT34 motif of *Xenopus* APE1 is sufficient to activate ATM kinase activity in vitro (Fig. 4F). Taken together, these data suggest that APE1 can directly stimulate ATM kinase in vitro and that the NT34 motif of APE1 is both required and sufficient for the stimulation.

### The lysine residues within APE1 NT34 motif are critical for APE1 dimer/oligomerization and its activation of ATM
To further investigate the underlying mechanism of APE1-mediated stimulation of ATM DDR, we tested whether APE1 protein forms dimers

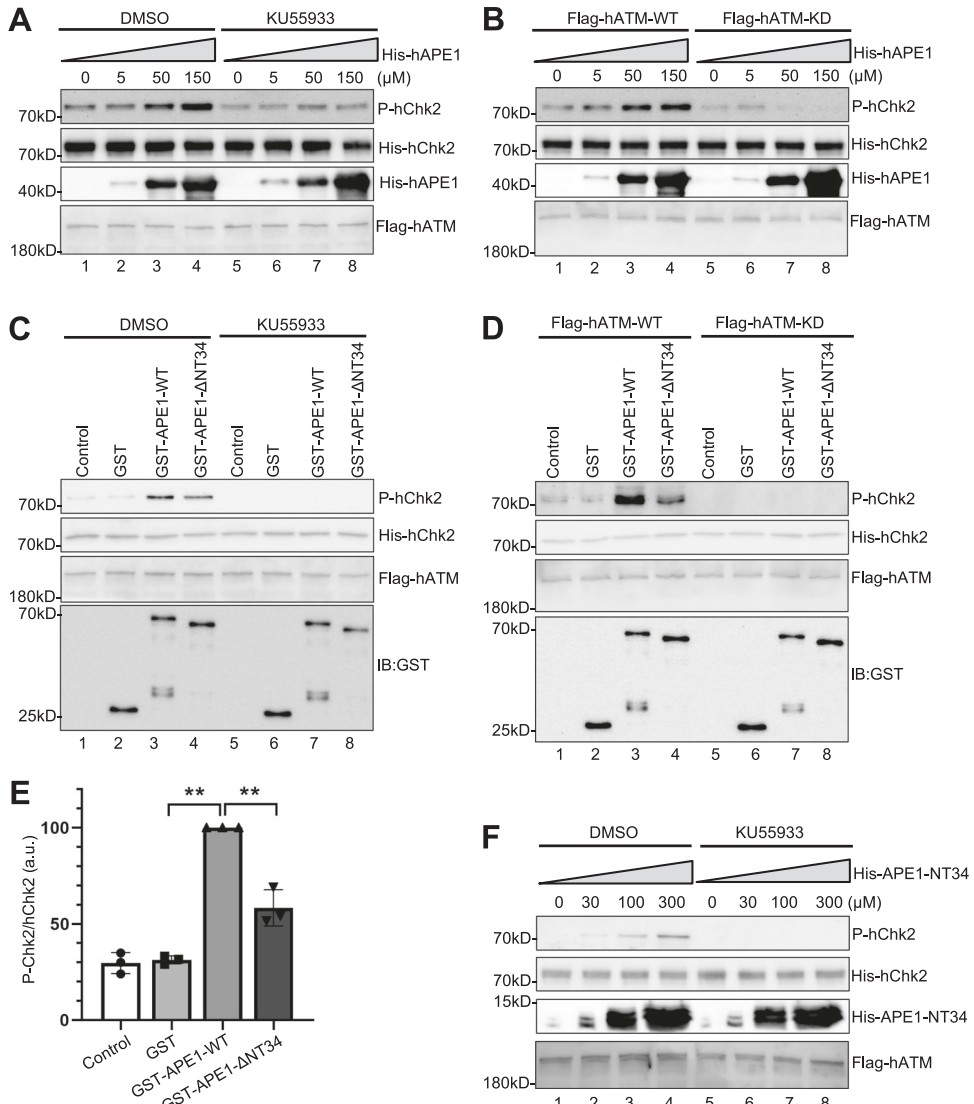

**Fig. 4 | APE1 directly stimulates ATM kinase activity in vitro. A** In vitro kinase assays were performed with different concentrations of His-tagged human APE1 (His-hAPE1) as indicated, Flag-tagged WT human ATM (Flag-hATM) and His-tagged kinase-deficient human Chk2 (His-hChk2), with DMSO or ATM inhibitor KU55933 (1 mM). ATM kinase activity was indicated by Chk2 phosphorylation (P-hChk2) via immunoblotting using anti-Chk2 T68 phosphorylation specific antibodies. **B** Kinase assays were performed with different concentrations of His-hAPE1 as indicated, WT or KD (kinase-dead) Flag-hATM and His-hChk2. ATM kinase activity was indicated by P-hChk2. **C** Kinase assays were performed with Control (PBS), GST, WT or ΔNT34 GST-*Xenopus* APE1, Flag-hATM and His-hChk2, with the addition of DMSO or KU55933 (1 mM). ATM kinase activity was indicated by P-hChk2. **D** Kinase assays

were performed with Control (PBS), GST, GST-*Xenopus* APE1 WT or ΔNT34, Flag-tagged WT or kinase-dead (KD) ATM and His-hChk2. ATM kinase activity was indicated by P-hChk2. **E** Quantification and statistical analysis of the ratio of P-hChk2 vs. hChk2 from Panel **C**. Data are presented as mean values ± SD. **$p$(GST vs GST-APE1-WT) = 0.0001; **$p$(GST-APE1-WT vs GST-APE1-ΔNT34) = 0.0016; two-tailed, unpaired t-test, $n = 3$. **F** Kinase assays were performed with different concentrations of His-tagged APE1 NT34 (His-APE1-NT34) and DMSO/KU55933 as indicated. Chk2 phosphorylation was examined via immunoblotting analysis. The data presented in Panel **A**–**D** and **F** are representative of three biological replicates. Source data are provided as a Source Data file.

and/or oligomers. Our GST pulldown experiments showed that GST-APE1, but not GST, associated with Myc-tagged APE1 (Myc-APE1) in in vitro interaction buffer (Fig. 5A). This association with Myc-APE1 was impaired in ΔNT34 GST-APE1 and NT34 GST-APE1, suggesting that the NT34 motif within APE1 is required but not sufficient for the interaction with full-length APE1 in trans (Fig. 5A). To test whether the NT34 motif of APE1 is sufficient for motif self-interaction, we performed GST pulldown experiment and found that GST-APE1-NT34 could pulldown His-APE1-NT34 (Fig. S5A). These data suggest that the NT34 motif within APE1 may mediate a head-to-head mode of APE1 intermolecular interactions.

Next, we sought to elucidate the molecular determinants within the APE1 NTD motif for APE1 dimerization/oligomerization and ATM activation. Although many efforts have been made, structural

information about APE1 N-terminal domain remains elusive[53,79]. AlphaFold failed to predict the structure of the N-terminal 33 amino acids of hAPE1 (homologous to *Xenopus* APE1 NT34) (Fig. 5B)[80]. However, APE1 NTD motif has several Lysine residues conserved in humans, frogs, and mice (i.e., K6, K7, K25, K26, and K33) (Fig. 5B), which may be important for APE1-induced ATM activation in the HSS system. To test this idea, we mutated the lysine to arginine individually or in combination in this region, and found that none of the KR substitutions in GST-APE1 including the 5KR mutation (K6R/K7R/K25R/K26R/K33R) affected the ectopically overexpressed APE1-induced ATM activation in the HSS (Fig. S5B and S5D). Next, we mutated the lysine residue into alanine individually or in combination (Fig. 5B and S5C). Our results showed that K25A/K26A/K33A (3KA) and K6A/K7A/K25A/K26A/K33A (5KA) GST-APE1 failed to

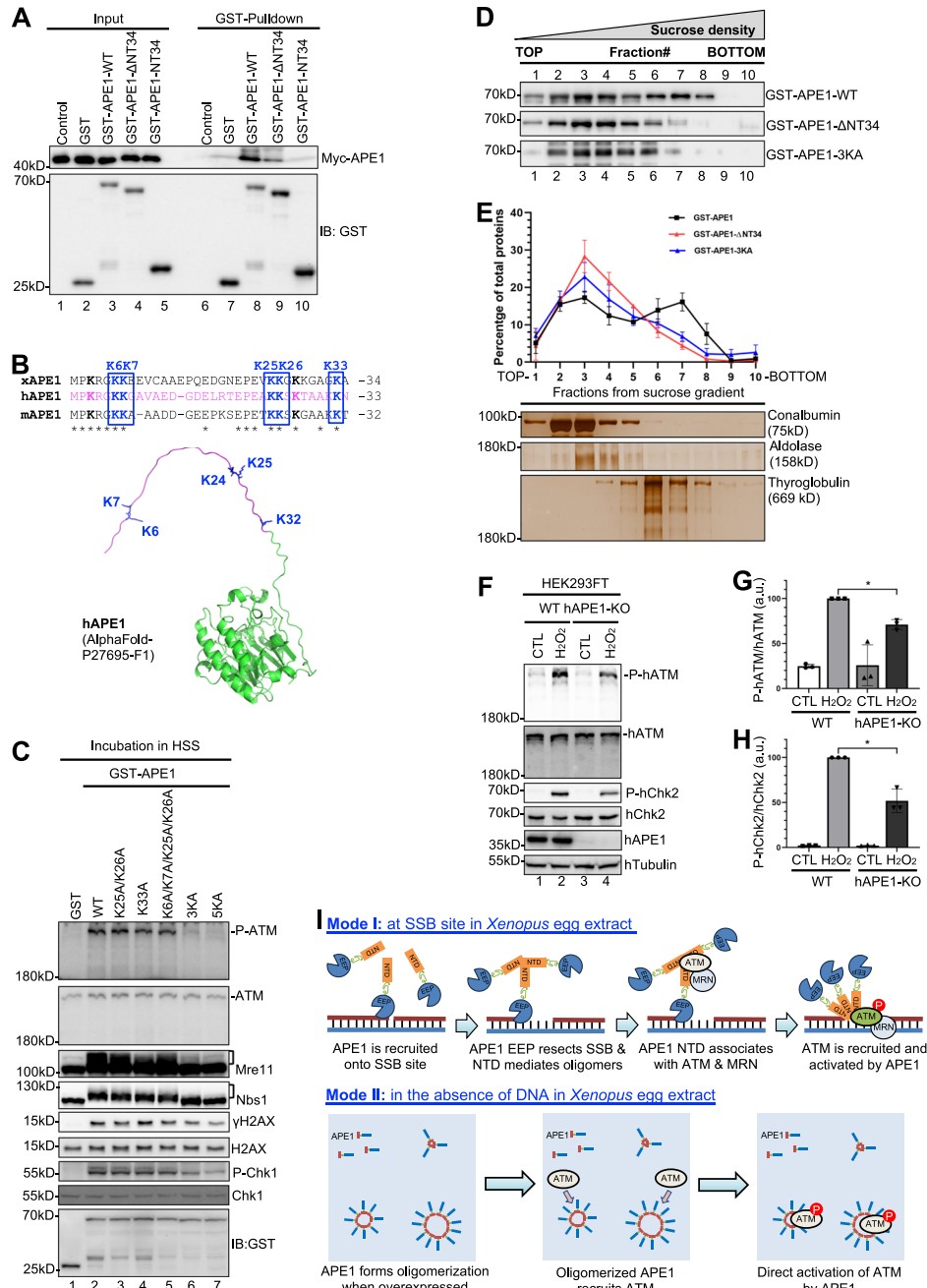

**Fig. 5 | Role of APE1 oligomerization in ATM activation and a conserved role of APE1 in ATM DDR in human cells. A** GST, WT, ΔNT34, or NT34 GST-APE1 was examined for interaction with Myc-APE1 in the interaction buffer. Input and GST-pulldown samples were examined via immunoblotting. **B** Amino acid alignment of APE1 N-terminal motifs from *Xenopus laevis* (xAPE1), humans (hAPE1), and mice (mAPE1). * indicates identical amino acids. The intrinsically disordered NT33 motif of hAPE1 was highlighted in magentas in a predicted structure from AlphaFold (AF-P27695-F1)[80]. **C** GST, WT, K25A/K26A, K33A, K6A/K7A/K25A/K26A, K25A/K26A/K33A (3KA), or K6A/K7A/K25A/K26A/K33A (5KA) GST-APE1 was added to HSS at a final concentration of 16 μM, and incubated for 30 min. Extracts were examined via immunoblotting. **D** Immunoblotting analysis of sucrose gradient fractions from WT, ΔNT34, 3KA GST-APE1. Each gradient was divided into 10 fractions, with fraction #1 representing the top and fraction #10 representing the bottom of the gradients. Proteins sediment within the gradient was based on their molecular mass. **E** Upper panel: Quantification results from experiments shown in Panel (D). Data

were presented as mean values ± SD; n = 3. Lower panel: Silver staining of sucrose gradient fractions from standard proteins: Conalbumin (75kD), Aldolase (158kD), Thyroglobulin (669kD). **F** APE1-knockout impaired $H_2O_2$-induced ATM DDR pathway in HEK-293FT cells. Total cell lysates were extracted and analyzed via immunoblotting analysis as indicated. **G–H** Quantification and statistical analysis of P-hATM vs hATM (G) and P-hChk2 vs hChk2 (H) from experiments shown in Panel (F). Data were presented as mean values ± SD. *$p = 0.0120$ (Panel G); *$p = 0.0234$ (Panel H); two-tailed, paired t-test, n = 3. (**I**) A working model for ATM activation by SSB and APE1. Mode I shows a mechanism of ATM signaling activation at SSB sites in *Xenopus* egg extracts. Mode II shows how ATM signaling is activated by APE1 overexpression in the absence of DNA in *Xenopus* egg extracts. See the main text for more details. The data presented in Panel **A, C, D** and **F** are representative of three biological replicates.} indicates non-specific bands. Source data are provided as a Source Data file.

activate ATM signaling in the HSS system (Fig. 5C). K25A/K26A, K33A, and K6K/K7AK25A/K26A mutant GST-APE1 still activated ATM activation, like WT GST-APE1 (Fig. 5C). GST pulldown experiment indicated that 3KA and 5KA GST-APE1 were deficient in interaction with Myc-APE1 (Fig. S5E). Furthermore, 3KA mutant significantly reduced the ability of GST-APE1 to stimulate ATM kinase activity as indicated by the phosphorylation of His-hChk2 in in vitro kinase assays (Fig. S5F, S5G). Additionally, WT but not 3KA Myc-APE1 rescued the SSB-induced phosphorylation of ATM, H2AX, Mre11, and Nbs1 in APE1-depleted HSS system (Fig. S5H). Together, these data suggest that the K25, K26, and K33 residues within the APE1 NTD motif are critical for APE1-APE1 interaction and its ability to activate ATM kinase.

To further validate whether and how APE1 forms dimer and/or oligomer, we performed a sucrose density gradient separation experiment measuring the estimated oligomeric status based on a given protein's molecular mass[81]. GST-APE1-WT was found in fraction #2 to #8, and peaked at fraction #3 and #7, supporting the notion that WT GST-APE1 forms different oligomers (Fig. 5D, E). However, ΔNT34 and 3KA GST-APE1 peaked at fraction #3 although they were found in fraction #2 to #6, suggesting that the K25, K26 and K33 residues within APE1 NTD motif are important for APE1 oligomerization (Fig. 5D, E). Three standard proteins (75kD Conalbumin, 158kD Aldolase, and 669kD Thyroglobulin) with different molecular weights were found in different fractions after sucrose density gradient separation (Fig. 5E). Monomeric GST-APE1-WT, GST-APE1-ΔNT34 and GST-APE1-3KA (approximately 70kD) should be detected in same/similar fractions as standard protein Conalbumin (75kD). We noted that ΔNT34 and 3KA GST-APE1 showed similar distribution pattern as Conalbumin, indicating that most ΔNT34 and 3KA GST-APE1 protein is likely monomer (Fig. 5D, E). Whereas Thyroglobulin (669kD) peaked at fraction #6, GST-APE1 peaked at fraction #3 and #7, suggesting that oligomeric APE1 can have a molecular weight higher than 669kD. These data suggest that the NTD motif of APE1 (especially the K25, K26 and K33 residues) is required for its dimer and/or oligomer in vitro.

**Conserved role of APE1 in the activation of ATM in human cells**
Although previous studies reported that APE1-knockout (KO) was lethal in mouse embryo and embryonic fibroblasts[82,83], two recent studies showed very mild phenotype of APE1-KO in human embryonic kidney cell line HEK293FT[84,85]. To explore the function of APE1 in ATM DDR pathway in mammalian cells, we chose hydrogen peroxide ($H_2O_2$) as a DNA damaging agent leading to oxidative stress-induced AP sites and SSBs[3,86], and found that ATM and Chk2 were phosphorylated after $H_2O_2$ treatment in WT HEK293FT cells, indicative of ATM DDR activation by $H_2O_2$-derived SSBs in the genome (Fig. 5F–H). Notably, the phosphorylation of ATM and Chk2 induced by $H_2O_2$ were significantly reduced in APE1-KO HEK293FT cells, suggesting APE1's important role in the activation of the ATM DDR in human cells (Fig. 5F–H). Furthermore, we found that purified His-hAPE1 protein stimulated robust ATM phosphorylation in a dose-dependent manner in an in vitro ATM DDR activation system using nuclear extracts isolated from human pancreatic cancer PANC1 cells (Fig. S5I). Of note, the phosphorylation of ATM induced by excess WT His-hAPE1-YFP in nuclear extracts was significantly reduced by the deletion of the NT33 motif in hAPE1 (Fig. S5J), which was consistent with our results from Xenopus system (Fig. 3E & 4C–E). To further test the conserved function of APE1 in the activation of ATM/ATR DDR during evolution, we examined whether purified His-hAPE1 protein can induce ATM and/or ATR DDR in the Xenopus HSS system. Our results showed that His-hAPE1 also triggered phosphorylation of ATM, Mre11, Chk1, and H2AX in the Xenopus HSS, albeit with a lower efficiency compared with GST-tagged Xenopus APE1 (Fig. S5K). These data suggest that APE1 is critical for activation of ATM DDR pathway in human cells, representing a conserved function of APE1 during evolution.

## Discussion

### SSB can induce and activate ATM kinase for SSB repair
The ATM kinase is a master regulator of the DDR, and it coordinates checkpoint activation, DNA repair, and metabolic changes in eukaryotic cells in response to DSBs and oxidative stress[9]. Loss of ATM activity in humans results in the pleiotropic neurodegeneration disorder ataxia-telangiectasia[23]. ATM exists in an inactive state in resting cells but can be activated by the MRN complex and DSBs[13,14,16]. In addition, oxidation of ATM activates its kinase activity independently of the MRN complex and DNA[30]. Thus, the SSB-induced ATM DDR activation is a distinct regulatory mechanism of ATM biology and regulation (Fig. 1 and S1). Previous studies have showed that ATM kinase is activated by SSB intermediates induced by $H_2O_2$ or MMS treatment, or presumptively accumulated SSBs in XRCC1-knockdown human primary fibroblasts to coordinate the replication of SSB-containing DNA and to prevent replication-born DSB formation[33,40–42]. Although ATM kinase is very likely activated by SSBs, these studies couldn't completely rule out the possibility that other types of DNA damage induced by $H_2O_2$, MMS, or XRCC1-knockdown are responsible for ATM activation. Here, we have established an SSB-induced ATM activation system in Xenopus HSS in which ATM-mediated DDR signaling is activated by a defined plasmid-based SSB structure containing 3'- and 5'-OH groups (Fig. 1 and S1). In this study, our data indicate that ATM kinase is recruited to defined SSB sites for ATM activation and ATM-mediated DDR activation in the cell-free Xenopus HSS system (Fig. 1 and S1). Our control experiments showing almost no detectable DSBs in our SSB preparation (Fig. S1A) and ~5 ng/μL of DSBs as a threshold for ATM activation in the HSS (Fig. S1B–S1C) ruled out the possibility of DSBs as a potential confounding factor in the SSB-induced ATM signaling in the HSS system.

Our previous studies have elucidated that majority of the defined SSB structure (at a concentration of as high as 75 ng/μL) was repaired in ~30 min in the HSS system[37,44,63], and that the repaired SSBs could be replicated upon the addition of NPE supplying S-CDK and DDK kinase activities[36,37]. Because there is no DNA replication elongation in the Xenopus HSS without NPE addition[49–51], we reason that the SSB structure in the HSS may not be repaired via replication-mediated DNA repair mechanisms. Due to the nature of 3'- and 5'-OH groups at our defined SSB structure, we don't think that the SSB is simply ligated together in the HSS either. How is the defined SSB structure repaired in the HSS system? Deep proteomics analysis has identified and quantified ~11,000 proteins in the Xenopus eggs[87], and Xenopus egg extracts contain DNA metabolism proteins such as DNA Ligases (e.g., DNA Lig I, III, and IV)[88,89], DNA polymerases (e.g., Pol alpha, beta, delta, and epsilon)[44,90–92], DNA endonucleases and exonucleases (e.g., APE1, APE2, Mre11, EXO1, and XPF-ERCC1)[19,37,63,93,94], and DDR kinase proteins (e.g., ATM and ATR)[21,37,61,90,95]. We have recently shown that APE1 and APE2 play critical roles in SSB repair via distinct SSB end resection in the 3'-5' direction[35–37,63], which modifies the SSBs into small ssDNA gap structures. Although conventional BER protein XRCC1 and Pol beta are dispensable for the repair of SSB plasmid in the HSS system[44], aphidicolin, inhibitor of Pol delta and epsilon, impaired the repair of SSB structure in HSS, suggesting potential contribution of Pol delta and epsilon to the DNA repair synthesis in the step of gap filling in SSB repair. A recent comprehensive review article on SSB repair also points out that defined SSB structures could be unwound by DNA helicase RECQ1 and resected by APE1/APE2 in the 3'−5' direction and that the subsequent ssDNA gap would be filled by DNA polymerases and ligated by DNA ligases[31]. Future studies are warranted to find out what exact DNA polymerases (Pol delta and/or epsilon?) and DNA ligases (Lig I and/or III?) are required for the repair of defined SSB structures in eukaryotic systems.

How is the SSB-induced ATM DDR coordinated with SSB repair in the HSS system? Our results suggest that ATM is recruited at SSB sites for phosphorylation and activation (Figs. 1K and 2I), and that SSB-

induced ATM signaling promotes SSB repair (Fig. 2A–D). How would the SSB-induced ATM contribute to SSB repair exactly? A previous study has shown that Chk2 forms a complex with XRCC1 and phosphorylates XRCC1 in vivo and in vitro at Thr284 to promote DNA repair[96]. It is possible that SSB-activated ATM kinase may phosphorylate Chk2 which in turn promotes the downstream DNA repair via XRCC1 phosphorylation. Because activated ATM can phosphorylate many substrate proteins involved in DNA metabolism[11], we can't exclude the possibility that other proteins involved in SSB repair can be phosphorylated and regulated by activated ATM at SSB sites[3,35]. It is important in future studies to identify additional components of the SSB repair pathway promoted by the ATM DDR pathway.

### Distinct regulatory mechanisms of ATM activation by APE1 in the presence of SSBs and in the absence of DNA in *Xenopus* egg extract

Previous studies have shown that ATM DDR signaling pathway is activated by DSBs through the MRN complex or by oxidative stress[4,9,17,97]. In the current study, our findings suggest two distinct regulatory mechanisms of ATM activation by APE1 in *Xenopus* egg extracts.

First, ATM-mediated DDR signaling is activated by APE1 at SSB sites in *Xenopus* egg extract (Mode I, Fig. 5I). We demonstrate that a defined SSB structure directly induces ATM pathway activation (indicated by phosphorylation of ATM, H2AX, Mre11, and Nbs1) in the *Xenopus* HSS system (Fig. 1). During the early stage of the SSB response, ATM and the MRN complex are recruited to SSB sites for ATM activation and SSB repair (Figs. 1–2). Importantly, APE1 is required for ATM DDR activation by SSB but not DSB structures (Fig. 2 and S2). Results of further mechanistic experimentations have elucidated that APE1 interacts and recruits ATM and MRN complex via its NTD motif to SSB sites, and that APE1 exonuclease activity is important to create the appropriate structures for ATM recruitment and activation (Fig. 2). Thus, ATM DDR can be activated by a defined SSB structure directly in an APE1-dependent manner. Therefore, our working model of how ATM is activated at SSB sites as follows (Mode I, Fig. 5I): (Step 1) APE1's EEP domain can sense and associate with SSB site[53,54,98,99]; (Step 2) APE1's NTD domain may mediate APE1 oligomerization, whereas APE1 can initiate the SSB end resection in the 3′–5′ direction via its EEP domain's exonuclease activity to generate a small ssDNA gap[63]; (Step 3) APE1's NTD domain associates with ATM and the MRN complex; and (Step 4) ATM kinase is recruited and activated by APE1 protein directly. Whether the conformation and/or biophysical property of APE1 monomer or oligomers at SSB site or converted gap structure are changed or transitioned warrants future studies for further clarifications.

Second, ATM-mediated DDR signaling is activated by ectopically overexpressed APE1 protein in the absence of DNA in *Xenopus* egg extract (Mode II, Fig. 5I). We recently reported that recombinant hAPE1 protein can form biomolecular condensates in vitro in an DNA/RNA-independent manner, and that ectopically overexpressed YFP-hAPE1 can be recruited to the nucleoli of human cancer cells in the absence of DNA damage to promote the ATR signaling[65]. Importantly, we have found in this study that excess addition of purified recombinant *Xenopus* APE1 protein can trigger the activation of ATM- and ATR-mediated DDR signaling in the absence of DNA in the *Xenopus* HSS system (Fig. 3), and that APE1 protein can directly stimulate ATM kinase activity in in vitro kinase assays (Fig. 4). Furthermore, the lysine residues within APE1 NTD motif are critical for APE1 dimerization/oligomerization and its stimulation of ATM kinase activity (Fig. 5 & S5). These data support a distinct mechanism of direct activation of ATM by overexpressed APE1 in the absence of DNA in *Xenopus* egg extract (Mode II, Fig. 5I): (Step 1) When overexpressed, high concentration of APE1 protein can form different species of APE1 dimers and/or oligomers in *Xenopus* egg extract, which is dependent on its NTD motif; (Step 2) Oligomerized APE1 can recruits ATM protein into APE1 oligomers with high local concentrations in the absence of DNA in

*Xenopus* egg extract; (Step 3) ATM kinase is directly activated by APE1 protein. Future experiments are warranted to better characterize the distinct features of APE1 oligomerization formation and disassembly by biophysical and molecular/cell biology approaches such as mass photometry technology, super-resolution microscopy analysis, and single-molecular technologies.

Our data using MRN depletion or Mre11 specific inhibitor suggest a critical role of MRN complex in the SSB-induced ATM activation in the HSS system (Fig. 1 and S1). We speculate several possible scenarios as possible underlying mechanisms: First, Mre11's nuclease activity may process the defined SSB structure in the 5′–3′ direction to enlarge the small ssDNA, which in turn facilitates ATM recruitment and activation. This is more or less reminiscent of the role of Mre11 nuclease activity in the 5′–3′ DSB end resection and ATM signaling[18,21,100]. Second, the MRN complex may promote the APE1-mediated ATM recruitment via yet to be characterized non-catalytic protein-protein interactions. As Rad50's ATP binding and DNA unwinding at DSB sites contribute to the DSB-induced ATM activation[16], we could not exclude this possibility in the context of SSB-induced ATM signaling. Nevertheless, further studies are needed to test these different scenarios in the future. In addition, it is of note that ATM kinase activity was stimulated by purified APE1 protein in in vitro kinase assays in the absence of MRN complex (Fig. 4). Previous studies have elucidated the stimulated kinase-substrate association as a possible mechanism for the direct activation of ATM kinase by the MRN complex in vitro[14,16]. We speculate that high-concentration of APE1 protein may promote the formation of LLPS in in vitro kinase assays, which subsequentially recruit and increase the local concentrations of ATM kinase and Chk2 substrate and/or promote their association. This potential mechanism of APE1-mediated ATM activation may bypass the need of the MRN complex in the absence of DNA damage.

Our data indicate that excess addition of WT but not 3KA mutant APE1 protein (16 μM) triggered the activation of ATM and ATR signaling in the absence of DNA in *Xenopus* HSS system (Figs. 3 and 5C). If compared to endogenous APE1 protein, how many folds of ectopically overexpressed recombinant APE1 protein was added to the HSS in order to trigger ATM signaling? We compared known concentrations of recombinant WT GST-APE1 protein to different volumes of HSS and found that the endogenous APE1 protein in the HSS was estimated as ~1.5 μM (Fig. S3G). This estimated endogenous APE1 concentration in the HSS is consistent with endogenous *Xenopus* APE1 in *Xenopus* eggs (~1.5 μM) and is in the similar range of human APE1 protein in human kidney cell HEK293T (~2.8 μM), as previously reported[64,87,101]. Thus, the concentration of 16 μM of recombinant APE1 ectopically over-expressed in the HSS triggering ATM signaling is about ~10-folds increase, compared with the estimated ~1.5 μM of endogenous APE1 in the HSS.

What is the potential role of the APE1 overexpression-induced ATM-mediated DDR signaling? APE1 is often found over-expressed in cancer cells compared with normal cells, and associated with poor overall survival in cancer patients[102–104]. However, there is no comprehensive investigation on the overexpression levels of APE1 protein in cancer tissue compared with non-malignant tissue. Interestingly, a recent study showed that transient APE1 overexpression (even ~2 folds increase compared with endogenous APE1) in normal human esophageal epithelial cells can lead to chromosomal instability, mutational signature 3 phenotype (e.g., C > T, T > C, C > A substitutions), and G2/M arrest[105]; however, it remains unclear whether this G2/M arrest by APE1 overexpression is dependent on ATM kinase. In addition, over-expressed APE1 may be translocated to different sub-cellular compartments where APE1 can activate ATM signaling. It has been demonstrated recently that ectopically overexpressed APE1 is translocated to the nucleoli of cancer cells but not normal cells to form biomolecular condensates and impairs the transcription of ribosomal DNA and results in S and/or G2/M checkpoint response; however, future experiments are needed to test whether such nucleolar

phenotype by APE1 overexpression is mediated through ATM DDR signaling[65,76]. Deficiencies of APE1 and ATM have been implicated in the increase of reactive oxygen species (ROS) in mitochondria[106,107], suggesting that both APE1 and ATM may be translocated to mitochondria to regulate oxidative stress response.

## Direct activation of the ATM and ATR kinase by APE1

APE1 is a multifunctional enzyme that plays a major role in base excision repair and DNA damage signaling[56,58]. Our recent studies indicate that APE1 plays an important role in SSB-induced activation of the ATR-Chk1 DDR pathway and in SSB repair through its 3′–5′ SSB end resection[63]. Overexpressed human APE1 can also assemble into biomolecular condensates in vitro and in nucleoli to promote the ATR DDR activation independent of its nuclease/redox functions[65]. It has been shown that APE1 inhibitor AR03 and APE1i III compromised the ATM/Chk2 DDR pathway induced by MMS in human embryonic kidney 293 T cells[42]. Strikingly, we found that addition of purified recombinant APE1 protein induces the activation of ATM DDR pathway in the absence of DNA in the HSS (Fig. 3). Furthermore, we show that recombinant APE1 protein can directly stimulate ATM kinase activity to phosphorylate its substrate Chk2 in vitro (Fig. 4). These data suggest that APE1 can form dimers/oligomers to increase local APE1 concentrations, which in turn promote ATM recruitment and activation.

It is intriguing that APE1 is a direct activator of both the ATM (this study) and ATR kinases[65]. ATM kinase activation requires the NTD motif of *Xenopus* APE1 especially the K25, K26, and K33 residues (Figs. 4–5 and S5), while ATR kinase activation requires the NT33 motif of human APE1 and the middle ATR Activation Domain (AAD) domain containing the critical W119 residue[65]. Whereas TopBP1 and ETAA1 have been identified as direct activator proteins of ATR kinase and ATR-mediated DDR signaling[90,108–113], the MRN complex is likely the most recognized direct activator of ATM kinase from literature[14,16]. Interesting, the MRN complex also plays an essential role in ATR activation and signaling via TopBP1 recruitment onto RPA-coated single-stranded DNA[48], suggesting that the MRN complex plays dual regulatory roles for both ATM and ATM DDR signaling via distinct mechanisms depending on the context. Thus, we propose that APE1 may serve as a molecular switch to turn on ATM and ATR by different mechanisms or different conformations.

Previous studies have shown different dependencies and crosstalk and regulations between ATM and ATR signaling. For example, ATM functions upstream of ATR signaling in DSB response[114,115], while ATR regulates ATM signaling in response to DNA replication stress[3,116]. At the SSB sites, APE1-mediated ATM activation may also function upstream of ATR signaling at least partially (Fig. 1F and S1D). The APE1-overexpression-induced ATM activation could be an upstream event of ATR DDR in the absence of DNA in the HSS as ATM inhibitor KU55933 compromised all ATM/ATR-dependent phosphorylation events, while ATR inhibitor VE-822 only impaired the phosphorylation of ATR-dependent events (Fig. 3C). We have demonstrated in our recent study that APE1 associates with ATR, ATRIP, and RPA to trigger the ATR-Chk1 DDR pathway activation in nuclear extract isolated from human cancer cells[63]. We reasoned that overexpressed APE1 protein may also associate with ATR protein in the *Xenopus* HSS to induce ATR-mediated DDR signaling, in addition to stimulating ATM DDR signaling (Fig. 3C).

Overall, our findings in this study reveal that SSBs can activate the ATM DDR pathway and that APE1 plays a direct role in this ATM activation.

## Methods

### Experimental procedures for egg extracts preparation in *Xenopus laevis*

The care and use of *Xenopus laevis* followed established protocols approved by the Institutional Animal Care and Use Committee at the University of North Carolina at Charlotte (IACUC Protocol-22-023). The

*Xenopus* HSS were prepared from eggs derived from female African clawed frogs (*Xenopus laevis* with age of ~4–12 years were purchased from NASCO), as described previously[45,49,50]. Immuno-depletion of target proteins in the HSS was performed with a similar procedure as previously described[37,61,90]. For instance, to deplete APE1 from HSS, 40 μL of HSS was incubated with 20 μL of ProteinA Sepharose beads, which was pre-incubated with 20 μL of anti-APE1 antiserum over night at 4 °C with constant rotating. Typically, 3-round incubation with antiserum coupled ProteinA Sepharose beads is required to ensure complete depletion in HSS. Anti-*Xenopus* APE1 or APE2 antibodies were described previously[61,63].

### SSB signaling technology in the *Xenopus laevis* HSS

The procedure of SSB signaling experiment in HSS system have been described previously[36,37]. Generally, 32 μL of HSS was supplemented with 8 μL of either control or SSB plasmid (see below section for details) to different final concentration (e.g., 40 ng/μL). After different incubation times at room temperature, the 5 μL of reaction mixture (i.e., HSS-plasmid mixture) was added with 50 μL sample buffer, which was examined via immunoblotting analysis. Various inhibitors targeting different pathways, KU55933 (Selleckchem, Cat#S1092), VE-822 (Selleckchem, Cat#S7102), NU7441 (Selleckchem, Cat#S2638), AR03 (Axonmedchem. Cat#axon2136), Mirin (Sigma, Cat#M9948) were purchased from commercially available vendors.

### Preparation of SSB, DSB plasmid and FAM-SSB structure

The control (CTL) plasmid, SSB plasmid and DSB plasmid were produced as described previously[36,37]. The plasmid pS derived from pUC19 (designated as CTL plasmid) contains single recognition site of SbfI-HF and single recognition site of Nt.BstNBI (at same location). The pS was digested by Nt. BstNBI to generate a defined SSB plasmid and following with CIP treatment to remove the 5′-P. The pS was digested by SbfI-HF to generate a defined DSB plasmid and following with CIP treatment to remove the 5′-P. Both of plasmids were purified from agarose via QIAquick gel extraction kit and then optionally purified by phenol–chloroform extraction.

### Recombinant DNA and proteins

Oligonucleotides used for plasmids construction are listed in Table S1. GST-tagged recombinant *Xenopus* APE1 and its truncations were generated as described previously[37,61,63]. Recombinant plasmid pET-28a-xAPE1-NT34 was prepared by PCR amino acids 1-34 of APE1 from pGEX-4T1-xAPE1 into pET-28a at BamHI and XhoI sites. Recombinant plasmid pET-28a-hChk2-D347A was prepared by PCR full-length of human Chk2-D347A from Hchk2-D347A into pET-28a at HindIII and XhoI sites. Hchk2 (kin-)-PUNI10 was a gift from Stephen Elledge (Addgene plasmid #41908; http://n2t.net/addgene:41908; RRID: Addgene_41908)[11]. pET28HIS-hAPE1 was a gift from Primo Schaer (Addgene plasmid #70757; http://n2t.net/addgene:70757; RRID: Addgene_70757)[117]. Various point mutant plasmids were prepared using QuikChange II XL site-directed mutagenesis kit (Agilent). GST- or His-tagged recombinant proteins were expressed in *E. coli* DE3/BL21 and purified following vendor's standard manual. Purified recombinant proteins were examined on SDS-PAGE gels with commassie blue staining and western blot. Myc-tagged recombinant proteins were generated by TNT SP6 Quick Coupled Transcription/Translation System (Promega) with recombinant pCS2+MT-derived plasmids according to vendor's protocol.

### Immunoblotting analysis and antibodies

Anti-*Xenopus* APE1 and APE2 custom antibodies were described previously[61,63]. Custom antibodies against ATM, Mre11, Nbs1, and RPA32 and were described previously[19,21,118]. Antibodies against ATM phosphorylation at Ser1981 (Rockland Immunochemicals, Cat#200-301-500), Chk1 (Santa Cruz Biotechnology, Cat#sc-8408), Chk1 phosphorylation at

Ser345 (Cell Signaling Technology, Cat#2348), Chk2 (Santa Cruz Biotechnology, Cat#sc-9064), Chk2 phosphorylation at Thr68 (Santa Cruz Biotechnology, Cat#sc-16297-R), Flag (Thermo Fisher, Cat#MA1-91878), GST (Santa Cruz Biotechnology, Cat#sc-138), H2AX (Cell Signaling Technology, Cat#7631 S), H2AX phosphorylation at Ser139 (Cell Signaling Technology, Cat#2577 S), H3 (Abcam, Cat#ab1791), His (Santa Cruz Biotechnology, Cat#sc-8036), Myc (Santa Cruz Biotechnology, Cat#sc-40), RPA32 phosphorylation at Ser33 (Bethyl Laboratories, Cat#A300-246A), and PCNA (Santa Cruz Biotechnology, Cat#sc-56) were purchased from commercially available vendors. Immunoblotting analysis was performed following procedures described previously[37,61,90].

### Analysis of DNA repair products from the HSS system

The protocol of SSB plasmid repair products isolation from the HSS system have been described previously with some modifications[36,63]. HSS was diluted by 3 volumes of ELB before inhibitors and plasmid DNA addition. After a 10-min incubation with inhibitors on ice, 1/4 HSS was mixed with SSB plasmid at final concentration of 40 ng/μL. After different time of incubation at room temperature, the mixture was resuspended in nuclease-free water, extracted with phenol–chloroform, and precipitated by ethanol with the presence of sodium acetate and glycogen. The resuspended and purified DNA repair products were examined on 1% agarose gel and stained with ethidium bromide.

### Plasmid DNA-bound fraction isolation in the *Xenopus laevis* HSS system

For plasmid DNA-bound fraction isolation from HSS system, 50 μL HSS-plasmid mixture was diluted with 200 μL of egg lysis buffer (ELB, 250 mM sucrose, 2.5 mM $MgCl_2$, 50 mM KCl, 10 mM HEPES, pH 7.7) after room temperature incubation. Diluted mixture was spun through a 1 mL of sucrose cushion (0.9 M sucrose, 2.5 mM $MgCl_2$, 50 mM KCl, 10 mM HEPES, pH 7.7) at 8,288 x g for 15 min at 4 °C with a swinging bucket (Beckman Coulter rotor S241.5). After centrifugation, the supernatants were removed, and the DNA-bound protein factions were added with 30μL sample buffer and detected via immunoblotting.

### GST pulldown assays

For the GST pulldown experiments in HSS, 10 μg of GST or GST-tagged recombinant proteins were added into 50 μL HSS which was diluted with 150 μL Interaction buffer (100 mM NaCl, 5 mM $MgCl_2$, 10% (vol/vol) glycerol, 0.1% Nonidet P-40, 20 mM Tris–HCl at pH 8.0). After 4 h of incubation at 4 °C, 20 μL mixture was collected as Input and the remaining mixture was added by 100 μL Interaction Buffer supplemented with 25 μL washed glutathione beads. After overnight incubation at 4 °C, the supernatant of mixture was removed, and the beads were washed by 500 μL Interaction Buffer three times. After removing the buffer remained in beads by needles, sample buffer was added to the beads. Then, the input and bead-bound fractions were examined via immunoblotting as indicated. For the GST pulldown experiments in interaction buffer, 10 μg of GST or GST-tagged recombinant proteins, and 10 μg His-tagged proteins, or 20 μL of TNT SP6 reactions containing various Myc-tagged recombinant proteins were added to 150 μL of Interaction Buffer (100 mM NaCl, 5 mM $MgCl_2$, 10% (vol/vol) glycerol, 0.1% Nonidet P-40, 20 mM Tris–HCl at pH 8.0). The following procedure was as same as the GST pulldown experiments in HSS.

### Cell culture and preparation of cell nuclear extracts and total cell lysates

HEK293 (ATCC #CRL-1573), WT/APE1-KO HEK293FT (Kind gift from Dr. Bruce Demple and Dr. Dmitry Zharkov), and PANC1 (ATCC #CRL-1469) cell lines were cultured in Dulbecco's modified Eagle's medium supplemented with 10% fetal bovine serum, and penicillin (100 U/mL) and streptomycin (100 μg/mL). Cells were maintained at 37 °C in a humidified incubator with 5% $CO_2$. The nuclear extracts were prepared as previously described with some modifications[63]. The cells were washed

with phosphate-buffered saline (PBS) and resuspended in Solution A (20 mM Tris–HCl pH 7.4, 10 mM NaCl, 3 mM $MgCl_2$). After incubation on ice for 15 min, NP-40 was added to a final concentration of 0.5%. The extracts were vortexed and centrifuged to separate permeabilized nuclei from cytoplasmic fraction. These nuclei were recovered and lysed with Lysis Buffer A (20 mM Tris–HCl pH 8.0, 150 mM NaCl, 2 mM EDTA, 0.5% NP-40, 0.5 mM $Na_3VO_4$, 5 mM NaF, 5 μg/mL of Aprotinin and 10 μg/mL of Leupeptin), then centrifuged at 14,007 x g for 30 min to get the nuclear extracts ready to use. After $H_2O_2$ treatments, the cells were recovered and washed with PBS, and then resuspended in Lysis Buffer A. The total cell lysates were isolated by centrifugation at 14,007 x g for 30 min, as recently described[65].

### In vitro ATM kinase assay

In vitro ATM kinase assay and related recombinant proteins purification were performed as previous described with some modifications[14,16,77]. Briefly, WT and Kinase dead (KD) ATM were made by transient transfection of expression constructs into HEK293 cells, and Flag-tagged ATM was pulled down by Anti-Flag M2 Magnetic Beads (Sigma Cat#M8823). Expression constructs for Flag-ATM-WT (pcDNA3.1(+)Flag-His-ATM wt) and Flag-ATM-KD (pcDNA3.1(+)Flag-His-ATM kd) were gifts from Michael Kastan (Addgene plasmid #31985; http://n2t.net/addgene:31985; RRID: Addgene_31985. Addgene plasmid #31986; http://n2t.net/addgene:31986; RRID: Addgene_31986)[119]. His-tagged Chk2 proteins used as ATM substrates were expressed in *E. coli* DE3 and purified following vendor's standard manual. The concentration of proteins was measured by Bradford assay (Bio-Rad) and determined by quantitation of protein using standard Coomassie-stained SDS-PAGE gels and western blotting. ATM kinase assays were performed in two steps: ATM beads were first incubated in stimulating buffer (50 mM HEPES, 100 mM NaCl, 1 mM $MnCl_2$ and 2 mM DTT) at 30 °C for 60 min. Then mix 3 μg of Chk2, 5 μg of APE1 and enough buffer A to bring the total volume to 20 μL, followed by adding 20 μL ATM beads and 20 μL kinase buffer (50 mM HEPES, pH 7.5, 50 mM KCl, 5 mM $MgCl_2$, 1 mM dithiothreitol (DTT), 1 mM ATP, and 10% glycerol). Incubate the mixture at 30 °C for 30 min. ATM kinase activity was determined by phosphorylation level of its substrate Chk2, which was detected by Western blot.

### Sucrose gradient separation

Sucrose gradients (2 mL) were formed by layering 250 μL each of 5, 10, 15, 20, 25, 30, 35, and 40% sucrose in Interaction Buffer (100 mM NaCl, 5 mM $MgCl_2$, 0.1% Nonidet P-40, 20 mM Tris–HCl at pH 8.0) and incubating for 1 h at room temperature and then 4 h at 4 °C. Samples were loaded onto the gradients and centrifuged at 77,100 x g for 16 h at 4 °C in a Beckman Coulter rotor TLS-55. 200 μL fractions were collected by pipetting from the top of the gradient. For the best resolution, the pipette tip should be placed against the wall of the tube and touching the gradients' surface. Proteins in each fraction are precipitated with trichloroacetic acid (TCA) and analyzed by immunoblotting analysis. Molecular size standards were purchased from Cytiva #28403842 and analyzed by SDS-PAGE and silver staining.

### Quantification and statistical analysis

The data presented are representative of three biological replicates unless otherwise specified. Graphical representation and statistical analysis were performed using GraphPad Prism (version 9). For significance analysis, t test with $p$ values was used to evaluate the difference between samples and noted in respective figure legends. For western blot and DNA agarose gel results, intensity analysis was performed using Image Lab software (Bio-Rad, version 6.0.1) with standard settings. Significance is denoted by asterisks in each figure: *$p < 0.05$; **$p < 0.01$; ns, no significance. Error bars represent the standard deviation (SD) for three independent experiments, unless otherwise indicated.

**Reporting summary**

Further information on research design is available in the Nature Portfolio Reporting Summary linked to this article.

## Data availability

All data are available in the main text or the Supplementary Information. The original Western blot data and quantifications of blot bands and associated statistical analysis data are provided in the Source Data file with this paper. Requests for the materials generated in this study should be directed to the corresponding author (S.Y.). Source data are provided with this paper.

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

## Acknowledgements

We thank Dr. Junya Tomida for assistance for the in vitro kinase assays. We are grateful to Dr. Bruce Demple, Dr. Dmitry O. Zharkov, Dr. Michael Kastan, Dr. Stephen Elledge, Dr. Primo Schaer, and Dr. Matthew Michael for reagents and cell lines. S.Y. thanks Zhe Li for reading and comments on the manuscript. The Yan lab is supported in part by the funds from University of North Carolina at Charlotte and grants from the NIH/NIEHS (R21ES032966 to S.Y.) and the NIH/NCI (R01CA225637 and R03CA270663 to S.Y.). The Lindsay lab is supported by North West Cancer Research grant (CR782 to H.D.L.).

## Author contributions

Conceptualization: H.Z., J.L., S.Y.; Formal analysis: H.Z., J.L., Z.Y., H.D.L., S.Y.; Funding acquisition: S.Y., H.D.L.; Investigation: H.Z., J.L., S.Y.; Methodology: H.Z., J.L., S.Y.; Project administration: S.Y.; Supervision: S.Y.; Visualization: H.Z., J.L., Z.Y., H.D.L., S.Y.; Writing – original draft: H.Z., S.Y.; Writing – review & editing: H.Z., Z.Y., H.D.L., S.Y.

## Competing interests

Authors declare that they have no competing interests.
