## [Peer Review File · Nature Communications]

Distinct regulation of ATM signaling by DNA single-strand breaks and APE1Editorial note: Parts of this Peer Review File have been redacted as indicated to maintain the confidentiality of unpublished data and to remove third-party material where no permission to publish could be obtained.

REVIEWER COMMENTS

Reviewer #1 (Remarks to the Author):

Strategy and rationale:

The study by Zhao et al examines a role for the SSB repair factor APE1 in activation of the ATM protein kinase. The main strategy employed is to incubate a plasmid containing a site-specific SSB in frog egg extracts and then assess ATM kinase activity using its own auto-phosphorylation on S1981 as well as phosphorylation of H2AX as readouts. Previous work from the corresponding author has shown that the SSB repair factor APE1 activates the ATM-related kinase ATR in this system and thus the focus of the current study was to determine if APE1 also activates ATM.

Data:

The goal of Figure 1 is to show that SSBs can indeed activate ATM in egg extract. In Fig1A we see that the SSB plasmid can stimulate ATM kinase more so than the control, except at the highest concentration where both control and SSB-containing plasmids stimulate ATM to very similar degrees. The authors show in Fig S1A that the control plasmid prep is a mixture of (mostly) closed circular DNA with some nicked circular DNA present in the sample. The presence of nicked circular DNAs in the control likely explains why it can activate ATM at high concentration, however the authors make no mention of this and they probably should. The authors go on to examine NBS1 and MRE11 phosphorylation using mobility shift on SDS-PAGE as the readout (Figs 1C and 1D), and while it is clear that SSBs are inducing mobility shifts the usual control of using phosphatase to show that the shifts are due to phosphorylation is missing, but this is not central to the interpretation of the data. In Fig 1E we see that ATMi blocks ATM activation by SSBs, but neither ATRi nor DNA-PKi do so. However, there are no efficacy controls for ATRi/DNA-PKi shown. Next we see that loss of MRN reduces ATM activation by SSBs, but there is still some signaling happening. This

experiment needs to be quantified, and the same is true for the Mirin experiment in Fig 1G. It is a very important point if MRN is needed for ATM signaling by SSBs and the available data (just a single timepoint) make it difficult to determine what is happening. Next, we see a time-course for ATM and ATR signaling and it is noted that ATM is activated earlier than ATR. Lastly, a DNA binding assay shows that ATM and MRN are physically bound to both the control and SSB plasmids and that for early timepoints there is more ATM/MRN on the SSB template. At the 15-minute timepoint, however, it appears that ATM and MRN come off the DNA? This is a little strange as we see in Fig 1H that ATM signaling is happening at 15 minutes (and beyond), so does this mean that ATM remains active after disengaging from the DNA? The authors should comment on this. Also, do the ATM bands observed in the control samples at all timepoints represent DNA-bound ATM or material that somehow made its way through the sucrose cushion independent of DNA? A “no DNA” control sample would settle this. Overall, I feel this Figure makes the point that the SSB plasmid can activate ATM, but there are some things that need tightening up and/or better explanations.

In Figure 2 we see early that repair of SSBs requires ATM kinase activity, which is very interesting, but this line of inquiry is then dropped and the focus shifts to the role of APE1 in ATR activation. We see that depletion of APE1 compromises ATM signaling and that add-back of WT but not mutant forms of APE1 can rescue the defect. The same is true for ATM recruitment to SSB plasmid. These data make it clear that APE1 is important for ATM signaling. The authors also include a GST pull-down showing that APE1 binds ATM and MRN in solution, suggesting that a preformed complex containing APE1-MRN-ATM exists in the extract independent of the presence of SSBs. One interesting point in this Figure is that the D306A mutant, defective in endonuclease activity, cannot activate ATM nor recruit it to SSBs. Do we know if this mutant itself stably associates with SSBs? If it does, then we learn that SSB binding by APE1 is not sufficient for ATM activation, and thus this experiment should be included in a revision. One minor issue with this Figure is I’m not sure what the point of the MRE11 panel is in Fig 2I “extracts” as nothing seems to change across the various conditions. Is this the correct image?

In Fig 3 we see that adding a large excess of APE1 to the egg extract causes ectopic activation of ATM (and ATR). Interestingly, we see that ectopic ATR signaling is blocked by

ATMi, suggesting that ATM plays a role in ATR activation by APE1. Is this also true when APE1 levels are normal and SSBs are used to activate ATR? This is important to know as for other systems, e.g. DSBs, ATR signaling is absolutely dependent on ATM but for stalled replication forks it is not. This would be a nice experiment to include in a revision. Also, can the D306A mutant ectopically activate ATM? This would be nice to know to get better resolution on the role of APE1's nuclease activity in ATM signaling.

Fig 4 uses in vitro kinase assays to examine how APE1 activates ATM. The authors use purified Chk2, APE1, and ATM for these experiments but they only show gels for purified Chk2 and APE1, but not ATM. They should show the ATM gel too so we can assess the level of purity. These data show that inclusion of APE1 indeed stimulates ATM in vitro, and thus supports the notion that APE1 acts directly on ATM and not through MRN. One minor issue is for Fig 4D why is there a band for P-Chk2 in lane 1? This should be blank as there are no ATM activators present in the sample, perhaps the authors can comment on this.

Figure 5 shows data suggesting that some lysine residues in the NT34 region of APE1 are important for oligomerization and for ATM activation. The data support the conclusion that APE1 interacts with itself and can form large multimers. The data also support a role for the lysines in NT34 in promoting multimerization and ATM signaling, however the only ATM assay used is the overexpression assay so it would be nice to see if the 5KA mutant fails to rescue an APE-1 depleted extract containing SSBs, which in my opinion is the more physiologically relevant assay. Lastly, the labeling on the y-axis in Fig 5E is messed up.

Overall evaluation:

This is an important study showing that APE1 can activate ATM kinase and I support its publication in Nature Communications. I do think a few additional experiments would strengthen the paper considerably and I list them below. I also think some important points in this study could be clarified in a more meaningful way and those are also listed below.

Additional experiments:

1. Can the D306A mutant bind to SSBs? Also, can this mutant ectopically activate ATM when

added in excess to extracts lacking SSBs?

2. Is ATM required for ATR signaling by SSBs, as it is during ectopic activation? For this simply add DMSO/ATMi to extracts containing control or SSB plasmids and assess ATR signaling.

3. Can the 5KA APE1 mutant rescue ATM activation in APE1-depleted extracts containing SSBs?

4. is ATM activity required for APE1 recruitment to SSBs?

Clarifications:

1. The authors should discuss why the control and SSB plasmids behave similarly in an ATM activation assay when used at high concentration.

2. Quantify the experiments shown in Figs 1F and 1G. Also, what do the authors think about the role of MRN in ATM activation by SSBs? The paper presents data that are seemingly in conflict with one another as we see that MRN depletion or MRE11i attenuates ATM signaling at SSBs but we also see in in vitro kinase assays that APE1 can activate ATM in the absence of MRN. What do the authors think MRN is doing in this system?

3. The authors should address the findings that ATM comes off DNA by 15 minutes in their system yet remains active for much longer.

4. Show the gel for purified ATM, if available.

Reviewer #2 (Remarks to the Author):

The Yan group follow up on their recent findings that showed APE1 recruiting ATRIP in response to DNA damage and assembles into condensates to drive the ATP-Chk1 pathway. Here, they find that ATM interacts with Ape1 in response to a single strand break. APE1 is again shown to form condensates and this process is presented as a mechanism to trap ATM

and DNA. More surprisingly, these functions are shown to occur in the absence of DNA when Ape1 is exogenously added to the lysate. They map the interaction site on Ape1 to the disordered N-terminal 34 amino acids, which is also responsible for the condensate forming properties. This reviewer appreciates the extensive amount of work, the careful concentration and time dependent changes in DNA response, and the clever amalgamation of ideas and principles. I have a few minor experimental issues listed below. Unfortunately, from a mechanistic standpoint, the findings and the model does not make a lot of sense. I fear that the findings are largely driven by artifacts of the experimental system. If I am mistaken, I would gladly like to read a response from the reviewers.

Major concerns

1. The authors state that the HSS fraction is devoid of DNA and thus the responses measured must be driven by the plasmid added. This seems like a straightforward interpretation and I agree with the assessment based on prior experimental evidence. The authors then go onto state that HSS is devoid of an replication activity and thus the ssDNA break remains intact. Can the authors describe what proteins are available in the HSS? I see PCNA is a component as they use it as a marker in their western blots. What about ligases? Polymerases? Endo- and Exo-nucleases. It would be helpful to have a better introduction to the system for a non-Xenopus reader. This is essential to understanding and interpreting the results.
2. The authors then go onto state that when Ape1 identifies the ssDNA break and process it, the DNA gets repaired and restored as shown in Fig. 2A. How is this possible if the system is devoid of DNA replication properties? Maybe I'm missing something here, but the two statements made by the authors appear to be contradictory.
3. The authors also present this as a clean system to interpret the results. But, there is significant stimulation of ATM phosphorylation even in the CTRL samples. For example, see the 40 and 60 ng/ul in Figures 1A-D. Yes, there is an increase upon addition of HSS. But, this begs the question as to why there is more than a 50% induction of these responses in the CTRL samples.
4. Western blots are also notoriously variable, and the entire paper is based on largely non-quantitative assessments of such data. This is worrisome, but to be fair, the qualitative interpretation does reasonably follow their interpretation.
5. In addition, the condensate experiment and the oligomerization properties are a stretch.

Yes, the IDP region in the N-terminus of Ape1 does appear to have some aggregation prone properties, but these need to be biophysically investigated. At a minimum, start with a simple assessment of what APE1 looks like in solution using cleaner biophysical tools (maybe Mass Photometry). The authors do have access to purified proteins and should be able to do these experiments.

My biggest concerns with the paper: Where is the specificity in the proposed mechanism? How can Ape1 randomly activate a DDR response in the absence and presence of a break? The final model invokes a DNA independent phosphorylation of ATM, and triggering of the MRE11 activity in the absence of DNA damage, what is the functional role for this phenomenon? This goes back to my original concern: what else is present in this HSS? My second concern is the addition of exogenous APE1 to drive condensate formation. What are the endogenous levels of Ape1 in the HSS1 fraction? What is 16 μ M Ape1 in terms of a driving force in terms of mass action? Is this a 2-fold excess or a 1000-fold excess? Finally, why does Ape1 trigger, the ATRIP, ATR, and even the RPA-Chk1 pathway and all together in the absence of DNA?

More realistically, are we missing some other cellular response all together? Can the increase in phosphorylation of the checkpoint proteins be attributed to some other immunogenic response instead?

In summary, the non-specific activation of DDR proposed by the authors are not in agreement with the first half of the paper that suggests that such changes are attributable to ssDNA breaks. Thus, I find the conclusions of the manuscript no suitable for publication.

Minor concerns

1. The title should read "Enhancement of ATM signaling by single-strand breaks and APE1".
2. Data in Fig 1H needs to be repeated and re-quantitated. How does anyone draw boxes around those bands. If this is their representative image, how are the error bars so small?
3. Please clarify the model in Figure 5I. A detailed interpretation and functional significance of the two phenomena needs to be discussed and a valid interpretation should be presented. The current discussion is largely a rehash of the results.
4. The word 'activated by' should be removed from the ms. The data suggests enhancement more than specific activation.
5. Page 11, line 304: 'stimulate' not 'stimulates'.

REVIEWER COMMENTS

Reviewer #1 (Remarks to the Author):

Strategy and rationale:

The study by Zhao et al examines a role for the SSB repair factor APE1 in activation of the ATM protein kinase. The main strategy employed is to incubate a plasmid containing a site-specific SSB in frog egg extracts and then assess ATM kinase activity using its own auto-phosphorylation on S1981 as well as phosphorylation of H2AX as readouts. Previous work from the corresponding author has shown that the SSB repair factor APE1 activates the ATM-related kinase ATR in this system and thus the focus of the current study was to determine if APE1 also activates ATM.

Data: The goal of Figure 1 is to show that SSBs can indeed activate ATM in egg extract. In Fig1A we see that the SSB plasmid can stimulate ATM kinase more so than the control, except at the highest concentration where both control and SSB-containing plasmids stimulate ATM to very similar degrees. The authors show in Fig S1A that the control plasmid prep is a mixture of (mostly) closed circular DNA with some nicked circular DNA present in the sample. The presence of nicked circular DNAs in the control likely explains why it can activate ATM at high concentration, however the authors make no mention of this and they probably should.

Response: Thanks for the suggestion. We have made changes in the text in Line 146-152 on Page 6 in the first paragraph of RESULTS, as shown below:

"We found that the SSB plasmid, but not CTL plasmid, at the concentrations of 20 ng/μL (~12 nM) and 40 ng/μL triggered robust ATM phosphorylation and γH2AX in the HSS system (Fig. 1A). It was noticed that ATM phosphorylation and γH2AX were also elevated to some extent by CTL plasmid at a higher concentration (i.e., 80 ng/μL) (Fig. 1A), likely due to contaminating SSB in the preparation of CTL plasmid (Fig. S1A). The quality and purity of SSB and DSB structures was confirmed by agarose gel electrophoresis and ethidium bromide staining (Fig. S1A)."

The authors go on to examine NBS1 and MRE11 phosphorylation using mobility shift on SDS-PAGE as the readout (Figs 1C and 1D), and while it is clear that SSBs are inducing mobility shifts the usual control of using phosphatase to show that the shifts are due to phosphorylation is missing, but this is not central to the interpretation of the data.

Response: Thank you for the suggestion. We performed additional experiment and found that the SSB-induced Mre11/Nbs1 mobility shifts were compromised by Lambda Phosphatase treatment (shown in new Fig. 1E and also here). We also added the description of this observation and made minor changes in Line 173-Line 178 on Page 7 (as shown here):

"To confirm the mobility shifts of Mre11 and Nbs1 induced by SSB plasmid were due to phosphorylation, we pre-treated the HSS with Lambda Phosphatase and found that the SSB-induced Mre11/Nbs1 mobility shifts were largely compromised (Fig. 1E). This observation suggests that SSB but not CTL plasmid triggers Mre11 and Nbs1 phosphorylation in the HSS system (indicated by their mobility shifts on gel). As expected, the SSB-induced ATM phosphorylation and γH2AX were also impaired by Lambda Phosphatase (Fig. 1E)."

New Fig. 1E. The SSB-induced mobility shifts of Mre11 and Nbs1 was compromised by Lambda Phosphatase in HSS.

In Fig 1E we see that ATMi blocks ATM activation by SSBs, but neither ATRi nor DNA-PKi do so.

However, there are no efficacy controls for ATRi/DNA-PKi shown.

Response: Thank you for your comment.

For the question of KU55933 as ATM kinase inhibitor, we added two references in our revised manuscript (Hickson et al., 2004, *Cancer Res*; You et al., 2007, *Nature Cell Biol*).

ATR kinase inhibitor VE-822 have been widely utilized as small molecule inhibitor for ATR DDR signaling. We added two previously published studies using VE-822 as ATR inhibitor in our revised manuscript (Fokas et al., 2012, *Cell Death Dis*; Lin et al., 2018, *Nucleic Acids Res*). Chk1 phosphorylation and RPA32 phosphorylation have been used as indicate of ATR kinase activation. We performed additional experiment

and found that VE-822 pretreatment in the HSS compromised SSB-induced Chk1 phosphorylation and RPA32 phosphorylation (New **Fig. S1D** and also here).

After screening of chromenone libraries, NU7441 was first identified as a highly potent and selective DNA-PKcs inhibitor (Leahy et al., 2004, *Bioorg Med Chem Lett*). Our previously published study has shown NU744 as a potent inhibitor of DNA-PK kinase in *Xenopus* egg extracts (Taylor et al., 2009, *Nucleic Acids Res*). A recent comprehensive review article also highlights NU7441 as the extensively utilized inhibitor of DNA-PKcs (Matsumoto, 2022, *Int J Mol Sci*). Whereas DNA-PKcs is not the focus of this study, we added these three publications as references for NU7441 in our revised manuscript.

With the newly added references and new data, we have modified below text in the revised manuscript **Line 178-Line 188 on Page 7**:

"To further determine whether the SSB-induced phosphorylation of ATM, H2AX, Nbs1 and Mre11 is dependent on ATM kinase in the HSS, we pre-incubated KU55933 (ATM kinase specific inhibitor)^{1,2}, VE-822 (ATR kinase specific inhibitor)^{3,4}, or NU7441 (DNA-PK kinase specific inhibitor)⁵⁻⁷ in the HSS prior to the addition of SSB or CTL plasmid (Fig. 1F). We found that KU55933 but neither VE-822 nor NU7441 compromised the SSB-induced phosphorylation of ATM, Nbs1, Mre11 and H2AX, suggesting that the defined SSB structure can induce ATM-dependent DDR signaling in the HSS system (Fig. 1F). Additionally, VE-822 pretreatment compromised the SSB-induced Chk1 phosphorylation at Ser344 (designated as P-Chk1, homologous to human Chk1 at Ser345) and RPA32 phosphorylation at Ser33 (designated as P-RPA32) (Fig. S1D). It was also noticed that KU55933 partially decreased the SSB-induced Chk1 and RPA32 phosphorylation (Fig. S1D)."

Next we see that loss of MRN reduces ATM activation by SSBs, but there is still some signaling happening. This experiment needs to be quantified, and the same is true for the Mirin experiment in Fig 1G. It is a very important point if MRN is needed for ATM signaling by SSBs and the available data (just a single timepoint) make it difficult to determine what is happening.

New Fig. S1D. The SSB-induced Chk1 and RPA32 phosphorylation was compromised by pretreatment of VE-822 in the HSS system.

Response: Thanks for reviewer's suggestion. The results shown in Fig. 1G are representative IB analysis of ATM signaling analysis comparing Mre11 depletion with mock-depletion (**Fig. 1G**). With another two replicates, we quantified the intensities of P-ATM vs total ATM, and formed statistical analysis, showing that the SSB-induced ATM phosphorylation was significantly reduced in Mre11-depleted HSS, compared with Mock-depleted HSS (**Fig. 1H**). In our revised manuscript, we moved the Mirin experiment results to **Fig. S1E**, and added quantification and statistical analysis of P-ATM/ATM from three experiments (Shown in **Fig. S1F**).

We performed the Mre11 depletion and Mirin addition experiments at the timepoint of 15-min incubation in the HSS system (**Fig. 1G-1H and S1E-S1F**). As suggested, we also conducted the Mre11 depletion experiment at 30-min timepoint (**Additional Fig. 1 (For reviewer's view only, not for publication)**), and found that the SSB-induced ATM phosphorylation was decreased after 30-min incubation in the HSS system, similar to our observation at 15-min timepoint. Because it seems that the 30-min experiment did not provide additional perspective on the role of Mre11 for SSB-induced ATM signaling, we decided not to include this 30-min experiment results in our revised manuscript.

Accordingly, we made some minor changes in the text in **Line 192-Line 199 on Page 7**:

"We found that the SSB-induced ATM phosphorylation and γH2AX were compromised when Mre11 was immunodepleted from the HSS (Fig. 1G-1H). The majority of Nbs1 was co-depleted from the HSS with anti-Mre11 antibodies, suggesting that the MRN complex is important for the activation of the SSB-induced ATM DDR (Fig. 1G-1H). Consistently, the addition of Mirin impaired SSB-induced phosphorylation of ATM, H2AX and Mre11 in the HSS system, suggesting that the

[figure redacted]

Additional Fig. 1 (For reviewer's view only, not for publication): 30-min timepoint of Mre11 depletion experiment. CTL or SSB plasmid was added to Mock- or Mre11-depleted HSS, followed by 30-min incubation and IB analysis.

exonuclease activity of Mre11 is important for the SSB-induced ATM DDR (Fig. S1E-S1F). Additionally, we observed similar results when Nbs1 was immunodepleted from the HSS with anti-Nbs1 antibodies (Fig. S1G)."

Next, we see a time-course for ATM and ATR signaling and it is noted that ATM is activated earlier than ATR. Lastly, a DNA binding assay shows that ATM and MRN are physically bound to both the control and SSB plasmids and that for early timepoints there is more ATM/MRN on the SSB template. At the 15-minute timepoint, however, it appears that ATM and MRN come off the DNA? This is a little strange as we see in Fig 1H that ATM signaling is happening at 15 minutes (and beyond), so does this mean that ATM remains active after disengaging from the DNA? The authors should comment on this.

Response: Thanks for the question. We have added a short description and commented on this in the text in Line 224-Line 229 on Page 8:

"Intriguingly, DNA-bound ATM and Mre11 at SSB sites were decreased at 15-min timepoint, while phosphorylated ATM was observed after 10-min incubation in total egg extracts (Fig. 1I-1K). We speculate that after phosphorylation and activation at SSB sites, ATM and Mre11 may disengage from the DNA into extracts and remain the phosphorylated and activated status. Our observations here reminded us of a previous study showing that ATM and Mre11 are recruited to DSB sites for activation and phosphorylated ATM and Mre11 dissociate from DNA in Xenopus egg extracts⁸."

Also, do the ATM bands observed in the control samples at all timepoints represent DNA-bound ATM or material that somehow made its way through the sucrose cushion independent of DNA? A "no DNA" control sample would settle this.

Response: Thank you for the suggestion. The DNA bound experiment with the "no DNA" control was performed and the results showed that very tiny to almost no detectable ATM, APE1 or Histone3 was in DNA-bound fraction from "No DNA" condition and that ATM and APE1 were recruited to CTL plasmid to some extent, although ATM and APE1 were highly enriched at SSB plasmid (Fig. S1H and also here). These observations support ATM signals observed in the DNA-fractions from CTL plasmid represent truly DNA-bound ATM proteins. We added one sentence on this control experiment in the text in Line 222-Line 223 on Page 8:

"Our control experiment showed that almost no ATM or APE1 was detected in DNA-bound fractions when no DNA was added in the HSS system (Fig. S1H)."

Overall, I feel this Figure makes the point that the SSB plasmid can activate ATM, but there are some things that need tightening up and/or better explanations.

In Figure 2 we see early that repair of SSBs requires ATM kinase activity, which is very interesting, but this line of inquiry is then dropped and the focus shifts to the role of APE1 in ATR activation. We see that depletion of APE1 compromises ATM signaling and that add-back of WT but not mutant forms of APE1 can rescue the defect. The same is true for ATM recruitment to SSB plasmid. These data make it clear that APE1 is important for ATM signaling. The authors also include a

GST pull-down showing that APE1 binds ATM and MRN in solution, suggesting that a preformed complex containing APE1-MRN-ATM exists in the extract independent of the presence of SSBs. One interesting point in this Figure is that the D306A mutant, defective in endonuclease activity, cannot activate ATM nor recruit it to SSBs. Do we know if this mutant itself stably associates with SSBs? If it does, then we learn that SSB binding by APE1 is not sufficient for ATM activation, and thus this experiment should be included in a revision.

Response: We appreciate the comments. However, we did include this data in our original submission and showed that D306A could stably associate with SSB plasmid (Lane 6, APE1 blot in the "DNA-bound" Panel, Fig. 2I). In contrast, we found that D306A mutation decreased the recruitment of ATM to SSB plasmid (Lane 6, ATM blot in the "DNA-bound" Panel, Fig. 2I). We recently showed that WT APE1 protein can resect the defined SSB into 1-3nt gap structure *in vitro* but D306A mutant APE1 failed to do so (Lin et al., 2020, *Nucleic Acids Res*). Our interpretation of the data is that once being recruited to SSB sites, APE1 may use its exonuclease activity to process the SSB in the 3' to 5' direction into small gap structures, which are the DNA structures to facilitate or enhance ATM recruitment. Based on these observations, we updated our working model indicating that the exonuclease activity of APE1 is required for ATM recruitment (Fig. 5I: Mode I).

One minor issue with this Figure is I'm not sure what the point of the MRE11 panel is in Fig 2I "extracts" as nothing seems to change across the various conditions. Is this the correct image?

Response: Thank you for pointing out this. This Mre11 blot was done with 8% PAGE gel which couldn't separate phosphorylated Mre11 from unmodified Mre11. We redid this experiment using 5% PAGE gel and replaced the Mre11 blot in "Extracts" panel showing the Mre11 mobility shifts (Lane 2 and Lane 4, Fig. 2I and also shown here).

In Fig 3 we see that adding a large excess of APE1 to the egg extract causes ectopic activation of ATM (and ATR). Interestingly, we see that ectopic ATR signaling is blocked by ATMi, suggesting that ATM plays a role in ATR activation by APE1. Is this also true when APE1 levels are normal and SSBs are used to activate ATR? This is important to know as for other systems, e.g. DSBs, ATR signaling is absolutely dependent on ATM but for stalled replication forks it is not. This would be a nice experiment to include in a revision.

Response: We agreed with this suggestion and performed such as suggested experiment. In the revised manuscript, our data showed that ATM inhibitor KU55933 compromised but not completely abolished the SSB-induced ATR activation (indicated by Chk1 phosphorylation and RPA32 phosphorylation) in the

HSS system (**Fig. S1D**). This result indicates that ATM signaling may also play an upstream role for SSB-induced ATR signaling in the HSS system when APE1 levels are normal.

Also, can the D306A mutant ectopically activate ATM? This would be nice to know to get better resolution on the role of APE1's nuclease activity in ATM signaling.

Response: As suggested, we added purified D306A mutant GST-APE1 protein in HSS and then detected the activation of ATM signaling pathway (new **Fig. S3F**). We found that ectopically overexpressed D306A GST-APE1 in the HSS activated the phosphorylation of ATM and H2AX and enhanced the mobility shifts of Mre11 and Nbs1, in a similar fashion as WT GST-APE1 control (new **Fig. S3F**). These observations indicate that APE1's exonuclease activity is dispensable for ATM activation induced by ectopically overexpressed APE1 protein in the HSS in the absence of DNA.

Fig 4 uses in vitro kinase assays to examine how APE1 activates ATM. The authors use purified Chk2, APE1, and ATM for these experiments but they only show gels for purified Chk2 and APE1, but not ATM. They should show the ATM gel too so we can assess the level of purity. These data show that inclusion of APE1 indeed stimulates ATM in vitro, and thus supports the notion that APE1 acts directly on ATM and not through MRN.

Response: Thank you for the suggestion. We added the coomassie staining of purified Flag-hATM on PAGE gel in new **Fig. S4B** in our revised manuscript (also shown here). As expected, purified Flag-hATM from cultured HEK293 cells are visible from PAGE gel, although there may be some ATM-interacting proteins co-purified (indicated by *). With the KU55933 inhibitor and ATM-KD control experiments in our in vitro kinase assays, we are confident that hChk2 phosphorylation stimulated by APE1 protein is mediated by ATM kinase itself but not by other potentially containing or co-purified ATM-interacting proteins.

One minor issue is for Fig 4D why is there a band for P-Chk2 in lane 1? This should be blank as there are no ATM activators present in the sample, perhaps the authors can comment on this.

Response: Thank you for the question. We repeated this experiment and replaced P-Chk2 blot in Fig. 4D.

Figure 5 shows data suggesting that some lysine residues in the NT34 region of APE1 are important for oligomerization and for ATM activation. The data support the conclusion that APE1 interacts with itself and can form large multimers. The data also support a role for the lysines in NT34 in promoting multimerization and ATM signaling, however the

New Fig. S3F. Excess D306A GST-APE1 activated

New Fig. S4B. Coomassie blue staining of Flag-hATM protein.

Fig. 4D with new "P-hChk2" blot.

only ATM assay used is the overexpression assay so it would be nice to see if the 5KA mutant fails to rescue an APE-1 depleted extract containing SSBs, which in my opinion is the more physiologically relevant assay. Lastly, the labeling on the y-axis in Fig 5E is messed up.

Response: We appreciated the suggestion. Our *in vitro* kinase assays showed that the ATM-mediated Chk2 phosphorylation was decreased by GST-APE1-3KA, compared to WT GST-APE1 (Fig. S5F-S5G), while excess addition of GST-APE1-3KA and GST-APE1-5KA failed to activate the ATM signaling in the HSS in the absence of DNA (Fig. 5C). We decided to test whether 3KA mutant APE1 fails to rescue the SSB-induced ATM signaling in the APE1-depleted HSS. As shown in new Fig. S5H (also shown here), WT but not 3KA Myc-APE1 protein rescued the SSB-induced phosphorylation of ATM, H2AX, Mre11, and Nbs1 in APE1-depleted HSS.

In addition, we also addressed issue of Y-axis labeling in Fig. 5E in our revised manuscript.

Overall evaluation:

This is an important study showing that APE1 can activate ATM kinase and I support its publication in Nature Communications. I do think a few additional experiments would strengthen the paper considerably and I list them below. I also think some important points in this study could be clarified in a more meaningful way and those are also listed below.

Additional experiments:

1. Can the D306A mutant bind to SSBs? Also, can this mutant ectopically activate ATM when added in excess to extracts lacking SSBs?

Response:

- In our original submission, we have showed that D306A APE1 could stably associate with SSB DNA (Lane 6, APE1 blot, "DNA-bound" Panel, Fig. 2I) whereas Δ NT34 compromised APE1 association with SSB. In contrast, D306A mutant could not rescue the recruitment of ATM and Mre11 to SSB plasmid ("DNA-bound" Panel, Fig. 2I). Furthermore, D306A-APE1 fails to activate ATM pathway in APE1-depleted HSS indicating that APE1's exonuclease activity is essential for ATM activation induced by SSB in HSS.
- As above described, we performed additional experiment and found that ectopically overexpression of D306A GST-APE1 still activated ATM signaling in the HSS in the absence of DNA (new Fig. S3F).

2. Is ATM required for ATR signaling by SSBs, as it is during ectopic activation? For this simply add DMSO/ATMi to extracts containing control or SSB plasmids and assess ATR signaling.

Response: As above described, our new experiment showed that ATM inhibitor KU5933 compromised, but not totally abolished, SSB-induced ATR activation in the HSS system (new Fig. S1D). This result indicated that SSB-induced ATM signaling is important for SSB-induced ATR signaling, the same as ectopically overexpressed APE1-induced DDR signaling pathways.

3. Can the 5KA APE1 mutant rescue ATM activation in APE1-depleted extracts containing SSBs?

Response: We have shown in our original submission that 3KA GST-APE1 (i.e., K25A/K26A/K33A) failed to active the ATM signaling in ectopic overexpress experiment in the

HSS (**Fig. 5C**) and decreased ATM-mediated Chk2 phosphorylation in in vitro kinase assays (**Fig. S5F-S5G**). We chose to test the role of 3KA APE1 in the SSB-induced ATM signaling activation in the HSS system, and found that WT but not 3KA Myc-APE1 rescued the SSB-induced ATM phosphorylation and γ H2AX as well as Mre11 and Nbs1 mobility shifts in APE1-depleted HSS (new **Fig. S5H**). This result indicates that at least three lysine residues within APE1 N-terminal motif are essential for SSB-induced ATM activation in the *Xenopus* HSS system. As 5KA mutant APE1 contains two more mutant in addition to the three lysine residues in 3KA mutant, we predict the 5KA APE1 mutant failed to rescue the SSB-induced ATM signaling in APE1-depleted HSS.

4. is ATM activity required for APE1 recruitment to SSBs?

Response: We appreciated the suggestion. We performed an additional experiment by pretreatment of HSS with KU55933 for SSB-bound experiment and found that ATM kinase inhibition by its inhibitor KU55933 had almost no noticeable effect on the recruitment of APE1 to SSB plasmid (New **Fig. S2G**). We added a short description of this experiment in the main text **Line 262-Line 265 on Page 9:**

"In contrast, ATM inhibitor KU55933 almost had almost no noticeable effect on the recruitment of APE1 to SSB sites (Fig. S2G), suggesting that APE1 association onto SSB sites is not regulated by ATM kinase as a potential feedback mechanism after its activation."

Clarifications:

1. The authors should discuss why the control and SSB plasmids behave similarly in an ATM activation assay when used at high concentration.

Response: As shown in our above response to earlier question, we have made changes in the text with more discussion in **Line 146-Line 152 on Page 6:**

"We found that the SSB plasmid, but not CTL plasmid, at the concentrations of 20 ng/ μ L (~12 nM) and 40 ng/ μ L triggered robust ATM phosphorylation and γ H2AX in the HSS system (Fig. 1A). It was noticed that ATM phosphorylation and γ H2AX were also elevated to some extent by CTL plasmid at a higher concentration (i.e., 80 ng/ μ L) (Fig. 1A), likely due to contaminating SSB in the preparation of CTL plasmid (Fig. S1A). The quality and purity of SSB and DSB structures was confirmed by agarose gel electrophoresis and ethidium bromide staining (Fig. S1A)."

2. Quantify the experiments shown in Figs 1F and 1G. Also, what do the authors think about the role of MRN in ATM activation by SSBs? The paper presents data that are seemingly in conflict with one another as we see that MRN depletion or MRE11i attenuates ATM signaling at SSBs but we also see in in vitro kinase assays that APE1 can activate ATM in the absence of MRN. What do the authors think MRN is doing in this system?

Response: Quantification results were added in revised manuscript (new **Fig. 1H** and **S1F**). For the potential mechanisms of MRN complex in the SSB-induced ATM signaling and in vitro ATM kinase assays, we have added additional discussions and clarifications in the text in **Line 549-Line 566 on Page 18**

" Our data using MRN depletion or Mre11 specific inhibitor suggest a critical role of MRN complex in the SSB-induced ATM activation in the HSS system (Fig. 1 and S1). We speculate

several possible scenarios as possible underlying mechanism: First, Mre11's nuclease activity may process the defined SSB structure in the 5'-3' direction to enlarge the small ssDNA, which in turn facilitates ATM recruitment and activation. This is more or less reminiscent of the role of Mre11 nuclease activity in the 5'-3' DSB end resection and ATM signaling^{2,9,10}. Second, the MRN complex may promote the APE1-mediated ATM recruitment via yet to be characterized non-catalytic protein-protein interactions. As Rad50's ATP binding and DNA unwinding at DSB sites contribute to the DSB-induced ATM activation¹¹, we could not exclude this possibility in the context of SSB-induced ATM signaling. Nevertheless, further studies are needed to test these different scenarios in the future. In addition, it is of note that ATM kinase activity was stimulated by purified APE1 protein in in vitro kinase assays in the absence of MRN complex (Fig. 4). Previous studies have elucidated the stimulated kinase-substrate association as a possible mechanism for the direct activation of ATM kinase by the MRN complex in vitro^{11,12}. We speculate that high-concentration of APE1 protein may promote the formation of LLPS in in vitro kinase assays, which subsequently recruit and increase the local concentrations of ATM kinase and Chk2 substrate and/or promote their association. This potential mechanism of APE1-mediated ATM activation may bypass the need of the MRN complex in the absence of DNA damage."

3. The authors should address the findings that ATM comes off DNA by 15 minutes in their system yet remains active for much longer.

Response: As stated in our earlier response, we added a short description and commented on this in the text in Line 224-Line 229 on Page 8:

"Intriguingly, DNA-bound ATM and Mre11 at SSB sites were decreased at 15-min timepoint, while phosphorylated ATM was observed after 10-min incubation in total egg extracts (Fig. 1I-1K). We speculate that after phosphorylation and activation at SSB sites, ATM and Mre11 may disengage from the DNA into extracts and remain the phosphorylated and activated status. Our observations here reminded us of a previous study showing that ATM and Mre11 are recruited to DSB sites for activation and phosphorylated ATM and Mre11 dissociate from DNA in *Xenopus* egg extracts⁸."

4. Show the gel for purified ATM, if available.

Response: Coomassie blue staining of Flag-hATM protein was added in revised manuscript (New Fig. S4B).

Reviewer #2 (Remarks to the Author):

The Yan group follow up on their recent findings that showed APE1 recruiting ATRIP in response to DNA damage and assembles into condensates to drive the ATP-Chk1 pathway. Here, they find that ATM interacts with Ape1 in response to a single strand break. APE1 is again shown to form condensates and this process is presented as a mechanism to trap ATM and DNA. More surprisingly, these functions are shown to occur in the absence of DNA when Ape1 is exogenously added to the lysate. They map the interaction site on Ape1 to the disordered N-terminal 34 amino acids, which is also responsible for the condensate forming properties. This reviewer appreciates the extensive amount of work, the careful concentration and time dependent changes in DNA response, and the clever amalgamation of ideas and principles. I have a few minor experimental issues listed below. Unfortunately, from a mechanistic standpoint, the findings and the model does not make a lot of sense. I fear that the findings are largely driven by artifacts of the experimental system. If I am mistaken, I would gladly like to read a response from the reviewers.

Response: We appreciate many suggestions and concerns from the reviewer. We have tried our best to perform additional experiments or to provide more clarifications or discussion in our revised manuscript. We also included four additional figures for your review only (not published in this paper). Please see below point by point response and we hope your concerns have been addressed.

Major concerns

1. The authors state that the HSS fraction is devoid of DNA and thus the responses measured must be driven by the plasmid added. This seems like a straightforward interpretation and I agree with the assessment based on prior experimental evidence. The authors then go onto state that HSS is devoid of an replication activity and thus the ssDNA break remains intact. Can the authors describe what proteins are available in the HSS? I see PCNA is a component as they use it as a marker in their western blots. What about ligases? Polymerases? Endo- and Exo-nucleases. It would be helpful to have a better introduction to the system for a non-Xenopus reader. This is essential to understanding and interpreting the results.

Response: Thank you for your suggestions. First, various *Xenopus* egg extracts including LSS, HSS, and NPE have been developed in the past and widely utilized in the studies of DNA repair and DDR pathways. As this has been published in studies from us (Cupello et al., 2016, PMID: 27160070) and many others, we show here a simplified diagram of how *Xenopus* egg extracts are prepared and implemented for experimental studies (**Additional Fig. 2 (For reviewer's view only, not for publication)**).

Yes, we added more description of the *Xenopus* egg extract system in the Introduction section
Line 82-98 on Page 4:

"Various
Xenopus laevis
egg extract
systems including
Low-Speed
Supernatant
(LSS), High-
Speed
Supernatant

[figure redacted]

Additional Fig. 2 (For reviewer's view only, not for publication): adapted from Fig. 2 and Fig. 3 in our previously published 2016 *Int J Dev Biol* paper (Cupello et al., 2016, PMID: 27160070).

(HSS), and Nucleo-Plasmic Extract (NPE) have been developed and widely utilized in structure and function studies of DNA repair (e.g., DNA inter-strand crosslink (ICL) repair and DSB repair) and DDR pathways (e.g., ATR and ATM)¹³⁻¹⁸. Briefly, eggs from female African clawed frogs are crushed after series of centrifugations at low speed (20,000g) and high speed (260,000g) to make the HSS, which contain most (if not all) proteins derived from eggs (e.g., replication/repair/DDR proteins) but not endogenous chromatin DNA, organelles (e.g., mitochondria and ribosomes), yolk, pigment granules, and membrane fractions^{14,19-21}. Purified plasmid DNA containing no apparent lesions (wild type, WT) or defined site-specific DNA damage (e.g., ICL, DSB, SSB) can be added to the HSS as exogenous DNA source for replication/repair/DDR analyses; however, plasmid DNA in the HSS system can potentially form pre-Replication Complex but can't continue DNA replication elongation until the addition of NPE which provides the replication elongation-needed S-CDK and DDK kinase activities^{19,20,22}. Several advantages of using HSS system include loss of function characterization (e.g., immunodepletion of target protein of interest or adding small molecule inhibitor) and gain of function or rescue experiment analyses (e.g., adding purified WT or mutant recombinant proteins at desired concentrations)^{14,19,20,22}.

Second, the DNA repair of our defined SSB structure in the HSS system have been published in our previous studies. Our previous studies have demonstrated that most SSB structure can be repaired within ~30 min in the HSS, and that repaired SSB can be replicated upon the addition of NPE supplying CDK and DDK kinase activity (**Additional Fig. 3 (For reviewer's view only, not for publication)**) (Lin et al., 2018, *Nucleic Acids Res*). We have shown that DNA nucleases APE1 and APE2 may resect the SSB structure in the 3'- and 5' direction to make ssDNA gap structure (Lin et al., 2018, *Nucleic Acids Res*; Lin et al., 2020, *Nucleic Acids Res*). Intriguingly, conventional BER protein XRCC1 and Pol beta are dispensable for SSB repair in the HSS system (Cupello et al., 2019, *Biochem J*). Thus, we added additional discussion in the Discussion section regarding how the SSB structure is repaired especially relevant to DNA ligases, DNA endo- and exo-nucleases, and DNA polymerases in **Line 471-494 on Page 15-16**:

"Our previous studies have elucidated that majority of the defined SSB structure (at concentration of as high as 75ng/ μ L) was repaired in ~30 min in the HSS system^{3,13,23}, and that the repaired SSBs could be replicated upon the addition of NPE supplying S-CDK and DDK kinase activities^{3,24}. Because there is no DNA replication elongation in the *Xenopus* HSS without NPE addition¹⁹⁻²¹, we reason that the

[figure redacted]

Additional Fig. 3 (For reviewer's view only, not for publication): shown in Fig. 2 in our previously published 2018 *Nucleic Acids Res* paper (Lin et al., 2018, PMID: 29361157).

SSB structure in the HSS may not be repaired via replication-mediated DNA repair mechanisms. Due to the nature of 3'- and 5'-OH groups at our defined SSB structure, we don't think that the SSB is simply ligated together in the HSS either. How is the defined SSB structure repaired in the HSS system? Deep proteomics analysis has identified and quantified ~11,000 proteins in the *Xenopus* eggs²⁵, and *Xenopus* egg extracts contain DNA metabolism proteins such as DNA Ligases (e.g., DNA Lig I, III, and IV)^{26,27}, DNA polymerases (e.g., Pol alpha, beta, delta, and epsilon)^{13,28-30}, DNA endonucleases and exonucleases (e.g., APE1, APE2, Mre11, EXO1, and XPF-ERCC1)^{3,7,23,31,32}, and DDR kinase proteins (e.g., ATM and ATR)^{2,3,28,33,34}. We have recently shown that APE1 and APE2 play critical roles in SSB repair via distinct SSB end resection in the 3'- and 5' direction^{3,23,24,35}, which makes the SSBs into small ssDNA gap structures. Although conventional BER protein XRCC1 and Pol beta are dispensable for the repair of SSB plasmid in the HSS system¹³, aphidicolin, inhibitor of Pol delta and epsilon, impaired the repair of SSB structure in HSS, suggesting potential contribution of Pol delta and epsilon to the DNA repair synthesis in the step of gap filling in SSB repair. A recent comprehensive review article on SSB repair also points out that defined SSB structures could be unwound by DNA helicase RECQ1 and resected by APE1/APE2 in the 3'-5' direction and that the subsequent ssDNA gap would be filled by DNA polymerases and ligated by DNA ligases³⁶. Future studies are warranted to find out what exact DNA polymerases (Pol delta and/or epsilon?) and DNA ligases (Lig I and/or III?) are required for the repair of defined SSB structures in eukaryotic systems."

2. The authors then go onto state that when Ape1 identifies the ssDNA break and process it, the DNA gets repaired and restored as shown in Fig. 2A. How is this possible if the system is devoid of DNA replication properties? Maybe I'm missing something here, but the two statements made by the authors appear to be contradictory.

Response: We appreciate the concern raised by the reviewer. However, we wanted to point out that the simplified working model of the repair of defined SSB structure in the HSS has been published (Hossain et al., 2019, PMID: 30110897). We adapted the Fig. 2 from our published paper, and also showed here of how SSB is repaired in the HSS (**Additional Fig. 4 (For reviewer's view only, not for publication)**): (Step 1) SSB end is sensed and processed by some DNA enzymes; (Step 2) SSB end resection is initiated by APE1 in the 3'-5' direction (Lin et al., 2020, *Nucleic Acids Res*); (Step 3) SSB end resection in the 3'-5' direction is continued by APE2 to trigger ATR DDR signaling (Lin et al., 2018, *Nucleic Acids Res*); (Step 4) SSB end resection is terminated by currently unknown mechanism; (Step 5) ssDNA gap is filled by DNA polymerases; and (Step 6) DNA nick is sealed by DNA ligases. As our focus in current manuscript

[figure redacted]

Additional Fig. 4 (For reviewer's view only, not for publication): adapted from Fig. 2 in our previously published 2019 *Int J Mol Sci* paper (Hossain et al., 2019, PMID: 30110897).

is how SSB is sensed by APE1 and how ATM is activated by SSB and APE1, we pay more attention on new insights into ATM recruitment and activation by APE1 at SSB sites (shown in **NEW Fig. 5I**). We indeed added more description and discussion of SSB repair in the Discussion section (please also see our response to the first question).

3. The authors also present this as a clean system to interpret the results. But, there is significant stimulation of ATM phosphorylation even in the CTL samples. For example, see the 40 and 60 ng/ul in Figures 1A-D. Yes, there is an increase upon addition of HSS. But, this begs the question as to why there is more than a 50% induction of these responses in the CTRL samples.

Response: Thanks for the suggestion. As shown in our response to first reviewer, our CTL plasmid may have some contaminated SSB and this may be the reason why ATM is also elevated to some extent at a high concentration (i.e., 80 ng/μL). We have made changes in the text in Line 146-Line 152 on Page 6 in the first paragraph of Results, as shown below:

"We found that the SSB plasmid, but not CTL plasmid, at the concentrations of 20 ng/μL (~12 nM) and 40 ng/μL triggered robust ATM phosphorylation and γH2AX in the HSS system (Fig. 1A). It was noticed that ATM phosphorylation and γH2AX were also elevated to some extent by CTL plasmid at a higher concentration (i.e., 80 ng/μL) (Fig. 1A), likely due to contaminating SSB in the preparation of CTL plasmid (Fig. S1A). The quality and purity of SSB and DSB structures was confirmed by agarose gel electrophoresis and ethidium bromide staining (Fig. S1A)."

4. Western blots are also notoriously variable, and the entire paper is based on largely non-quantitative assessments of such data. This is worrisome, but to be fair, the qualitative interpretation does reasonably follow their interpretation.

Response: We agree that the variability inherent in Western blots is acknowledged by researchers. However, this does not diminish their utility when employed judiciously alongside rigorous controls and normalization techniques. We also try our best to mitigate the potential adverse impact of this method by quantifying some of our blot results (e.g., **New Fig. 1H** is quantification and statistical analysis of data from **New Fig. 1G**). And as well known, western blotting remains the most widely used and pivotal method to date for protein detection in biological research because it offers crucial valuable insights into protein expression and dynamics, enriching our understanding of biological processes.

5. In addition, the condensate experiment and the oligomerization properties are a stretch. Yes, the IDP region in the N-terminus of Ape1 does appear to have some aggregation prone properties, but these need to be biophysically investigated. At a minimum, start with a simple assessment of what APE1 looks like in solution using cleaner biophysical tools (maybe Mass Photometry). The authors do have access to purified proteins and should be able to do these experiments.

Response: We appreciate the suggestion and potential importance and value of incorporating Mass Photometry experiments into our study. As our university doesn't have such equipment, we reached out to a core facility of another research institution which has a first-generation of Mass Photometry instrument (Refeyn Ltd., Oxford, UK). Therefore, we set up a trial experiment of Mass Photometry analysis using our purified recombinant WT GST-APE1 (MW: ~58kD) in a Tris buffer. WT GST-APE1 performed best at a concentration of ~125nM and gave a primary mass peak of ~49kD representing a ~15% error for a monomer and a secondary peak of ~149kD presumably a dimer. Some higher MW readings were either indicative of oligomerization or light aggregation. This preliminary observation of MP analysis is more or less consistent with our observations of APE1 self-interaction and dimer/oligomerization shown in **Fig. 5**; however, the higher MW species seem not quite satisfactory to our quality control. Because all our responses to comment will be

in a "peer review file" after the paper is published, we decided to omit the MP data in this point by point response and just described for reviewer's reference.

As shown in our recently published study of human APE1 in biomolecular condensation *in vitro*, the condensate formation requires APE1 protein to be at the range of micromolar concentrations (Li et al., 2022, *Nucleic Acids Res*). A dilemma is created between the nature of APE1's condensate formation requiring high concentration and the capture events by mass photometry requiring low concentration. Additionally, we consulted the technical support from the vender Refeyn and was informed that Refeyn just developed an add-on part and updated software which could titrate the highly concentrated recombinant proteins automatically for subsequent Mass Photometry analysis; however, this add-on might only be added to second-generation TWO^{MP} instrument and there is no expected timeline to add this add-on to any MP equipment yet.

We would like to add more biophysical approaches (such as Mass Photometry) of our WT and mutant APE1 proteins for condensate formation and disassembly in our future studies. Due to the current technical challenges and the lack of available instrument for such analysis, we would like to hold these more biophysical analyses in our future follow-up publications. We did at a sentence in the Discussion for future studies on this perspective at Line 544-547 on Page 18:

"Future experiments are warranted to better characterize the distinct features of APE1-mediated condensate formation and disassembly by biophysical and molecular/cell biology approaches such as mass photometry technology, super-resolution microscopy analysis, and single-molecular technologies."

My biggest concerns with the paper: Where is the specificity in the proposed mechanism? How can Ape1 randomly activate a DDR response in the absence and presence of a break? The final model invokes a DNA independent phosphorylation of ATM, and triggering of the MRE11 activity in the absence of DNA damage, what is the functional role for this phenomenon? This goes back to my original concern: what else is present in this HSS?

Response: Whereas we appreciate your concern, we disagree with the suggestion that "Ape1 randomly activate a DDR in the absence and presence of a break". To better illustrate the possible mechanisms from our data, we modified our working model in **New Fig. 5I** (and also here) and better explained and discussed the two modes of ATM DDR activation in the Discussion section Line 514-544 on Page 17-18:

Mode I: at SSB sites in *Xenopus* egg extract:

*"First, ATM-mediated DDR signaling is activated by APE1 at SSB sites in *Xenopus* egg extract (Mode I, Fig. 5I). We demonstrate that a defined SSB structure directly induces ATM pathway activation (indicated by phosphorylation of ATM, H2AX, Mre11, and Nbs1) in the*

Xenopus HSS system (Fig. 1). During early stage of the SSB response, ATM and the MRN complex are recruited to SSB sites for ATM activation and SSB repair (Fig. 1-2). Importantly, APE1 is required for ATM DDR activation by SSB but not DSB structures (Fig. 2 and S2). Results of further mechanistic experimentations have elucidated that APE1 interacts and recruits ATM and MRN complex via its NTD motif to SSB sites, and that APE1 exonuclease activity is important to create the appropriate structures for ATM recruitment and activation (Fig. 2). Thus, ATM DDR can be activated by a defined SSB structure directly in an APE1-dependent manner. Therefore, our working model of how ATM is activated at SSB sites as follows (Mode I, Fig. 5I): **(Step 1)** APE1 protein can form dimer and/or oligomer and is recruited onto SSB sites; **(Step 2)** APE1 may form molecular condensates at SSB sites and work with MRN complex to recruit ATM onto SSB sites; and **(Step 3)** ATM kinase is activated by APE1 protein directly at SSB sites."

Mode II: in the absence of DNA in *Xenopus* egg extract:

"Second, ATM-mediated DDR signaling is activated by ectopically overexpressed APE1 protein in the absence of DNA in *Xenopus* egg extract (Mode II, Fig. 5I). We recently reported that recombinant hAPE1 protein can form biomolecular condensates in vitro in an DNA/RNA-independent manner, and that ectopically overexpressed YFP-hAPE1 can be recruited to the nucleoli of human cancer cells in the absence of DNA damage to promote the ATR signaling³⁷. Importantly, we have found in this study that excess addition of purified recombinant *Xenopus* APE1 protein can trigger the activation of ATM- and ATR-mediated DDR signaling in the absence of DNA in the *Xenopus* HSS system (Fig. 3), and that APE1 protein can directly stimulates ATM kinase activity in in vitro kinase assays (Fig. 4). Furthermore, the lysine residues within APE1 NTD motif are critical for APE1 dimerization/oligomerization and its stimulation of ATM kinase activity (Fig. 5 & S5). These data support a distinct mechanism of direct activation of ATM by APE1 in the absence of DNA in *Xenopus* egg extract (Mode II, Fig. 5I): **(Step 1)** APE1 protein can form dimer and/or oligomer, which is dependent on its NTD motif; **(Step 2)** high concentration of APE1 protein may form molecular condensates in the absence of DNA in *Xenopus* egg extract, which in turn recruits ATM protein into APE1 condensates; **(Step 3)** ATM kinase is directly activated by APE1 protein."

What is the potential role of ATM activation induced by APE1 over-expression? To address this question, we have added a separate paragraph with appropriate references in Discussion Line 580-597 on Page 19:

"What is the potential role of the APE1-overexpression-induced ATM-mediated DDR signaling? APE1 is often found over-expressed in cancer cells compared with normal cells, and associated with poor overall survival in cancer patients³⁸⁻⁴⁰. However, there is no comprehensive investigation on the overexpression levels of APE1 protein in cancer tissue compared with non-malignant tissue. Interestingly, a recent study showed that transient APE1 overexpression (even ~2 folds increase compared with endogenous APE1) in normal human esophageal epithelial cells can lead to chromosomal instability, mutational signature 3 phenotype (e.g., C>T, T>C, C>A substitutions), and G2/M arrest⁴¹; however, it remains unclear whether this G2/M arrest by APE1 overexpression is dependent on ATM kinase. In addition, overexpressed APE1 may be translocated to different sub-cellular compartments where APE1 can activate ATM signaling. It has been demonstrated recently that ectopically overexpressed APE1 is translocated to the nucleoli of cancer cells but not normal cells to form biomolecular condensates and impairs the transcription of ribosomal DNA and results in S and/or G2/M checkpoint response; however, future experiments are needed to test whether such as nucleolar phenotype by APE1 overexpression is mediated through ATM

DDR signaling^{37,42}. Deficiencies of APE1 and ATM have been implicated in the increase of reactive oxygen species (ROS) in mitochondria^{43,44}, suggesting that both APE1 and ATM may be translocated to mitochondria to regulate oxidative stress response."

My second concern is the addition of exogenous APE1 to drive condensate formation. What are the endogenous levels of Ape1 in the HSS1 fraction? What is 16 μ M Ape1 in terms of a driving force in terms of mass action? Is this a 2-fold excess or a 1000-fold excess?

Response: Thanks for bringing up a good question. In our recently published studies (Lin et al., 2023, *eLife*), we discussed the estimated concentrations of endogenous APE1 protein from two setting: (1) the concentration of *Xenopus* APE1 protein in *Xenopus laevis* egg was estimated to be \sim 1.5 μ M (Wühr et al., 2014, *Current Biol*); and (2) the concentration of human APE1 protein in human kidney cells HEK293T was estimated to \sim 2.8 μ M (Wiśniewski et al., 2014, *Mol Cell Proteomics*). To address this question experimentally, we performed immunoblotting analysis of a titration of different volumes of HSS and different pmol of recombinant GST-APE1 protein on a same blot (**New Fig. S3G**). We found that the intensity of endogenous APE1 from 1 μ L of HSS in Lane 3 is close to the intensity of GST-APE1 protein between Lane 5 and Lane 6, suggesting that the concentration of endogenous APE1 in HSS is estimated as \sim 1.5 μ M. We understand that IB is not most accurate or best method to determine the concentration of endogenous APE1 in HSS, but our estimated \sim 1.5 μ M of endogenous APE1 in the HSS is consistent within the APE1 concentrations from *Xenopus* eggs and cultured human cells. Thus, the concentration of 16 μ M of recombinant APE1 ectopically over-expressed in the HSS for ATM signaling is about \sim 10 folds increase, compared with the estimated \sim 1.5 μ M of endogenous APE1 in the HSS.

We added this data and discussion in the Discussion **Line 570-578 on Page 18-19**:

"If compared to endogenous APE1 protein, how many folds of ectopically overexpressed recombinant APE1 protein was added to the HSS in order to trigger ATM signaling? We compared known concentrations of recombinant WT GST-APE1 protein to different volumes HSS and found that the endogenous APE1 protein in the HSS was estimated as \sim 1.5 μ M (**Fig. S3G**). This estimated endogenous APE1 concentration in the HSS is consistent with endogenous *Xenopus* APE1 in *Xenopus* eggs (\sim 1.5 μ M) and is in the similar range of human APE1 protein in human kidney cell HEK293T (\sim 2.8 μ M), as previously reported^{25,45,46}. Thus, the concentration of 16 μ M of recombinant APE1 ectopically over-expressed in the HSS triggering ATM signaling is about \sim 10 folds increase, compared with the estimated \sim 1.5 μ M of endogenous APE1 in the HSS."

Finally, why does Ape1 trigger, the ATRIP, ATR, and even the RPA-Chk1 pathway and all together in the absence of DNA?

Response: Thanks for your question. To better explain and discuss the ATM and ATR-mediated DDR signaling in the absence of DNA in the HSS system, we added two more paragraphs in Discussion **Line 613-637 Page 20**:

"It is intriguing that APE1 is a direct activator of both the ATM (this study) and ATR kinases³⁷. ATM kinase activation requires the NTD motif of *Xenopus* APE1 especially the K25, K26, and K33 residues (**Figs. 4-5 and S5**), while ATR kinase activation requires the NT33 motif of human APE1 and the middle ATR Activation Domain (AAD) domain containing the critical

W119 residue³⁷. Whereas TopBP1 and ETAA1 have been identified as direct activator proteins of ATR kinase and ATR-mediated DDR signaling^{28,47-52}, the MRN complex is likely the most recognized direct activator of ATM kinase from literature^{11,12}. Interesting, the MRN complex also plays an essential role in ATR activation and signaling via TopBP1 recruitment onto RPA-coated single-stranded DNA¹⁷, suggesting that the MRN complex plays dual regulatory roles for both ATM and ATM DDR signaling via distinct mechanisms depending on the context. Thus, we propose that APE1 may serve as a molecular switch to turn on ATM and ATR by different mechanisms or different conformations.

Previous studies have shown different dependencies and crosstalk and regulations between ATM and ATR signaling. For example, ATM functions upstream of ATR signaling in DSB response^{53,54}, while ATR regulates ATM signaling in response to DNA replication stress^{55,56}. At the SSB sites, APE1-mediated ATM activation may also function upstream of ATR signaling at least partially (Fig. 1F and S1D). The APE1-overexpression-induced ATM activation could be an upstream event of ATR DDR in the absence of DNA in the HSS as ATM inhibitor KU55933 compromised all ATM/ATR-dependent phosphorylation events, while ATR inhibitor VE-822 only impaired the phosphorylation of ATR-dependent events (Fig. 3C). We have demonstrated in our recent study that APE1 associates with ATR, ATRIP, and RPA to trigger the ATR-Chk1 DDR pathway activation in nuclear extract isolated from human cancer cells²³. We reasoned that overexpressed APE1 protein may also associate with ATR protein in the *Xenopus* HSS to induce ATR-mediated DDR signaling, in addition to stimulating ATM DDR signaling (Fig. 3C)."

More realistically, are we missing some other cellular response all together? Can the increase in phosphorylation of the checkpoint proteins be attributed to some other immunogenic response instead?

Response: Although we appreciate the reviewer's suggestion, we think that this study is focused on the role and mechanism of DDR signaling and immune response could be out of the scope of this study. But in the Introduction, we did add the role of APE1 in immune response with additional references (e.g., Guikema et al., 2007, J Exp Med; Oliveira et al., 2022, Font Immunol) **Line 103-105 on Page 5:**

"In addition to the well-characterized roles in base excision repair (BER), redox regulation, and immune response⁵⁷⁻⁶⁰, accumulating evidence support APE1's previously uncharacterized but critical roles in DDR signaling pathways."

In summary, the non-specific activation of DDR proposed by the authors are not in agreement with the first half of the paper that suggests that such changes are attributable to ssDNA breaks. Thus, I find the conclusions of the manuscript no suitable for publication.

Minor concerns

1. The title should read "Enhancement of ATM signaling by single-strand breaks and APE1".

Response: We appreciate the suggestion. However, we'd rather to change our title in our revised manuscript to "Distinct regulation of ATM signaling by DNA single-strand breaks and APE1".

2. Data in Fig 1H needs to be repeated and re-quantitated. How does anyone draw boxes around those bands. If this is their representative image, how are the error bars so small?

Response: We were not quite sure what the reviewer meant to suggest here. Of note, we can only run 14 samples on one SDS-PAGE gel so that the left half (14 samples) and right half (14 samples) of **Fig. 1I** were performed on two separate gels, and we included the samples from 30-min on both gel for comparison. One additional biological replicate of the experiment in **Fig. 1I** was performed and intensities of IB bands were re-quantified from three replicates (shown in **new Fig. 1J**).

3. Please clarify the model in Figure 5I. A detailed interpretation and functional significance of the two phenomena needs to be discussed and a valid interpretation should be presented. The current discussion is largely a rehash of the results.

Response: Thank you for the question. Please see our above response to a similar question.

4. The word 'activated by' should be removed from the ms. The data suggests enhancement more than specific activation.

Response: We agree with the reviewer that the Chk2 phosphorylation by ATM kinase was stimulated by the addition of WT GST-APE1 protein in *in vitro* kinase assays (Fig. 4). However, the phosphorylation events (e.g., ATM, H2AX, Mre11, Nbs1) induced by SSB plasmid in the HSS (Fig. 1 and 2) or by ectopically over-expressed recombinant APE1 protein in the HSS (Fig. 3) are the specific events and we have identified the distinct regulatory mechanisms mediated by APE1 protein. Although we appreciate your suggestion, with all due respect, we would like to keep "activated by" in the revised manuscript.

5. Page 11, line 304: 'stimulate' not 'stimulates'.

Response: Thank you for your careful review. This has been corrected and we have gone through the manuscript and checked all typos.

References:

- Hickson, I. *et al.* Identification and characterization of a novel and specific inhibitor of the ataxia-telangiectasia mutated kinase ATM. *Cancer Res* **64**, 9152-9159 (2004). <https://doi.org/10.1158/0008-5472.CAN-04-2727>
- You, Z., Bailis, J. M., Johnson, S. A., Dilworth, S. M. & Hunter, T. Rapid activation of ATM on DNA flanking double-strand breaks. *Nat Cell Biol* **9**, 1311-1318 (2007). <https://doi.org/10.1038/ncb1651>
- Lin, Y. *et al.* APE2 promotes DNA damage response pathway from a single-strand break. *Nucleic Acids Res* **46**, 2479-2494 (2018). <https://doi.org/10.1093/nar/gky020>
- Fokas, E. *et al.* Targeting ATR in vivo using the novel inhibitor VE-822 results in selective sensitization of pancreatic tumors to radiation. *Cell Death Dis* **3**, e441 (2012). <https://doi.org/10.1038/cddis.2012.181>
- Leahy, J. J. *et al.* Identification of a highly potent and selective DNA-dependent protein kinase (DNA-PK) inhibitor (NU7441) by screening of chromenone libraries. *Bioorg Med*

- Chem Lett* **14**, 6083-6087 (2004). <https://doi.org/10.1016/j.bmcl.2004.09.060>
- 6 Matsumoto, Y. Development and Evolution of DNA-Dependent Protein Kinase Inhibitors toward Cancer Therapy. *International journal of molecular sciences* **23** (2022). <https://doi.org/10.3390/ijms23084264>
- 7 Taylor, E. M. *et al.* The Mre11/Rad50/Nbs1 complex functions in resection-based DNA end joining in *Xenopus laevis*. *Nucleic Acids Res* **38**, 441-454 (2010). <https://doi.org/10.1093/nar/gkp905>
- 8 Di Virgilio, M., Ying, C. Y. & Gautier, J. PIKK-dependent phosphorylation of Mre11 induces MRN complex inactivation by disassembly from chromatin. *DNA Repair (Amst)* **8**, 1311-1320 (2009). <https://doi.org/10.1016/j.dnarep.2009.07.006>
- 9 Shibata, A. *et al.* DNA double-strand break repair pathway choice is directed by distinct MRE11 nuclease activities. *Mol Cell* **53**, 7-18 (2014). <https://doi.org/10.1016/j.molcel.2013.11.003>
- 10 Deng, Y., Guo, X., Ferguson, D. O. & Chang, S. Multiple roles for MRE11 at uncapped telomeres. *Nature* **460**, 914-918 (2009). <https://doi.org/10.1038/nature08196>
- 11 Lee, J. H. & Paull, T. T. ATM activation by DNA double-strand breaks through the Mre11-Rad50-Nbs1 complex. *Science* **308**, 551-554 (2005). <https://doi.org/10.1126/science.1108297>
- 12 Lee, J. H. & Paull, T. T. Direct activation of the ATM protein kinase by the Mre11/Rad50/Nbs1 complex. *Science* **304**, 93-96 (2004). <https://doi.org/10.1126/science.1091496>
- 13 Cupello, S., Lin, Y. & Yan, S. Distinct roles of XRCC1 in genome integrity in *Xenopus* egg extracts. *Biochem J* **476**, 3791-3804 (2019). <https://doi.org/10.1042/BCJ20190798>
- 14 Willis, J., DeStephanis, D., Patel, Y., Gowda, V. & Yan, S. Study of the DNA damage checkpoint using *Xenopus* egg extracts. *J Vis Exp*, e4449 (2012). <https://doi.org/10.3791/4449>
- 15 Raschle, M. *et al.* Mechanism of replication-coupled DNA interstrand crosslink repair. *Cell* **134**, 969-980 (2008). <https://doi.org/10.1016/j.cell.2008.08.030>
- 16 Ben-Yehoyada, M. *et al.* Checkpoint signaling from a single DNA interstrand crosslink. *Mol Cell* **35**, 704-715 (2009). <https://doi.org/10.1016/j.molcel.2009.08.014>
- 17 Duursma, A. M., Driscoll, R., Elias, J. E. & Cimprich, K. A. A role for the MRN complex in ATR activation via TOPBP1 recruitment. *Mol Cell* **50**, 116-122 (2013). <https://doi.org/10.1016/j.molcel.2013.03.006>
- 18 You, Z., Chahwan, C., Bailis, J., Hunter, T. & Russell, P. ATM activation and its recruitment to damaged DNA require binding to the C terminus of Nbs1. *Mol Cell Biol* **25**, 5363-5379 (2005). <https://doi.org/10.1128/MCB.25.13.5363-5379.2005>
- 19 Lebofsky, R., Takahashi, T. & Walter, J. C. DNA replication in nucleus-free *Xenopus* egg extracts. *Methods Mol Biol* **521**, 229-252 (2009). https://doi.org/10.1007/978-1-60327-815-7_13
- 20 Cupello, S., Richardson, C. & Yan, S. Cell-free *Xenopus* egg extracts for studying DNA

- damage response pathways. *The International journal of developmental biology* **60**, 229-236 (2016). <https://doi.org/10.1387/ijdb.160113sy>
- 21 Walter, J., Sun, L. & Newport, J. Regulated chromosomal DNA replication in the absence of a nucleus. *Mol Cell* **1**, 519-529 (1998). [https://doi.org/10.1016/s1097-2765\(00\)80052-0](https://doi.org/10.1016/s1097-2765(00)80052-0)
- 22 Walter, J. & Newport, J. Initiation of eukaryotic DNA replication: origin unwinding and sequential chromatin association of Cdc45, RPA, and DNA polymerase alpha. *Mol Cell* **5**, 617-627 (2000). [https://doi.org/10.1016/s1097-2765\(00\)80241-5](https://doi.org/10.1016/s1097-2765(00)80241-5)
- 23 Lin, Y. *et al.* APE1 senses DNA single-strand breaks for repair and signaling. *Nucleic Acids Res* **48**, 1925-1940 (2020). <https://doi.org/10.1093/nar/gkz1175>
- 24 Lin, Y., Ha, A. & Yan, S. Methods for Studying DNA Single-Strand Break Repair and Signaling in *Xenopus laevis* Egg Extracts. *Methods Mol Biol* **1999**, 161-172 (2019). https://doi.org/10.1007/978-1-4939-9500-4_9
- 25 Wuhr, M. *et al.* Deep proteomics of the *Xenopus laevis* egg using an mRNA-derived reference database. *Curr Biol* **24**, 1467-1475 (2014). <https://doi.org/10.1016/j.cub.2014.05.044>
- 26 Kumamoto, S. *et al.* HPF1-dependent PARP activation promotes LIG3-XRCC1-mediated backup pathway of Okazaki fragment ligation. *Nucleic Acids Res* **49**, 5003-5016 (2021). <https://doi.org/10.1093/nar/gkab269>
- 27 Graham, T. G., Walter, J. C. & Loparo, J. J. Two-Stage Synapsis of DNA Ends during Non-homologous End Joining. *Mol Cell* **61**, 850-858 (2016). <https://doi.org/10.1016/j.molcel.2016.02.010>
- 28 Yan, S. & Michael, W. M. TopBP1 and DNA polymerase-alpha directly recruit the 9-1-1 complex to stalled DNA replication forks. *J Cell Biol* **184**, 793-804 (2009). <https://doi.org/10.1083/jcb.200810185>
- 29 Waga, S., Masuda, T., Takisawa, H. & Sugino, A. DNA polymerase epsilon is required for coordinated and efficient chromosomal DNA replication in *Xenopus* egg extracts. *Proc Natl Acad Sci U S A* **98**, 4978-4983 (2001). <https://doi.org/10.1073/pnas.081088798>
- 30 Fukui, T. *et al.* Distinct roles of DNA polymerases delta and epsilon at the replication fork in *Xenopus* egg extracts. *Genes Cells* **9**, 179-191 (2004). <https://doi.org/10.1111/j.1365-2443.2004.00716.x>
- 31 Chen, X., Paudyal, S. C., Chin, R. I. & You, Z. PCNA promotes processive DNA end resection by Exo1. *Nucleic Acids Res* **41**, 9325-9338 (2013). <https://doi.org/10.1093/nar/gkt672>
- 32 Klein Douwel, D. *et al.* XPF-ERCC1 acts in Unhooking DNA interstrand crosslinks in cooperation with FANCD2 and FANCP/SLX4. *Mol Cell* **54**, 460-471 (2014). <https://doi.org/10.1016/j.molcel.2014.03.015>
- 33 Willis, J., Patel, Y., Lentz, B. L. & Yan, S. APE2 is required for ATR-Chk1 checkpoint activation in response to oxidative stress. *Proc Natl Acad Sci U S A* **110**, 10592-10597 (2013). <https://doi.org/10.1073/pnas.1301445110>
- 34 Robertson, K., Hensey, C. & Gautier, J. Isolation and characterization of *Xenopus* ATM (X-

- ATM): expression, localization, and complex formation during oogenesis and early development. *Oncogene* **18**, 7070-7079 (1999). <https://doi.org:10.1038/sj.onc.1203194>
- 35 Hossain, M. A., Lin, Y. & Yan, S. Single-Strand Break End Resection in Genome Integrity: Mechanism and Regulation by APE2. *International journal of molecular sciences* **19**, 2389 (2018). <https://doi.org:10.3390/ijms19082389>
- 36 Caldecott, K. W. DNA single-strand break repair and human genetic disease. *Trends Cell Biol* **32**, 733-745 (2022). <https://doi.org:10.1016/j.tcb.2022.04.010>
- 37 Li, J., Zhao, H., McMahon, A. & Yan, S. APE1 assembles biomolecular condensates to promote the ATR-Chk1 DNA damage response in nucleolus. *Nucleic Acids Res* **50**, 10503-10525 (2022). <https://doi.org:10.1093/nar/gkac853>
- 38 Jensen, K. A., Shi, X. & Yan, S. Genomic alterations and abnormal expression of APE2 in multiple cancers. *Scientific reports* **10**, 3758 (2020). <https://doi.org:10.1038/s41598-020-60656-5>
- 39 Kumar, S. *et al.* Role of apurinic/apyrimidinic nucleases in the regulation of homologous recombination in myeloma: mechanisms and translational significance. *Blood Cancer J* **8**, 92 (2018). <https://doi.org:10.1038/s41408-018-0129-9>
- 40 Yuan, C. L. *et al.* APE1 overexpression is associated with poor survival in patients with solid tumors: a meta-analysis. *Oncotarget* **8**, 59720-59728 (2017). <https://doi.org:10.18632/oncotarget.19814>
- 41 Kumar, S. *et al.* Elevated APE1 Dysregulates Homologous Recombination and Cell Cycle Driving Genomic Evolution, Tumorigenesis, and Chemoresistance in Esophageal Adenocarcinoma. *Gastroenterology* **165**, 357-373 (2023). <https://doi.org:10.1053/j.gastro.2023.04.035>
- 42 Li, J. & Yan, S. Molecular mechanisms of nucleolar DNA damage checkpoint response. *Trends Cell Biol* **33**, 361-364 (2023). <https://doi.org:10.1016/j.tcb.2023.02.003>
- 43 Li, M. X. *et al.* Human apurinic/apyrimidinic endonuclease 1 translocalizes to mitochondria after photodynamic therapy and protects cells from apoptosis. *Cancer Sci* **103**, 882-888 (2012). <https://doi.org:10.1111/j.1349-7006.2012.02239.x>
- 44 Lee, J. H. *et al.* ATM directs DNA damage responses and proteostasis via genetically separable pathways. *Science signaling* **11** (2018). <https://doi.org:10.1126/scisignal.aan5598>
- 45 Wisniewski, J. R., Hein, M. Y., Cox, J. & Mann, M. A "proteomic ruler" for protein copy number and concentration estimation without spike-in standards. *Molecular & cellular proteomics : MCP* **13**, 3497-3506 (2014). <https://doi.org:10.1074/mcp.M113.037309>
- 46 Lin, Y. *et al.* APE1 recruits ATRIP to ssDNA in an RPA-dependent and -independent manner to promote the ATR DNA damage response. *eLife* **12**, e82324 (2023). <https://doi.org:10.7554/eLife.82324>
- 47 Kumagai, A., Lee, J., Yoo, H. Y. & Dunphy, W. G. TopBP1 activates the ATR-ATRIP complex. *Cell* **124**, 943-955 (2006). <https://doi.org:10.1016/j.cell.2005.12.041>
- 48 Bass, T. E. *et al.* ETAA1 acts at stalled replication forks to maintain genome integrity. *Nat*

- Cell Biol* **18**, 1185-1195 (2016). <https://doi.org/10.1038/ncb3415>
- 49 Haahr, P. *et al.* Activation of the ATR kinase by the RPA-binding protein ETAA1. *Nat Cell Biol* **18**, 1196-1207 (2016). <https://doi.org/10.1038/ncb3422>
- 50 Lee, Y. C., Zhou, Q., Chen, J. & Yuan, J. RPA-Binding Protein ETAA1 Is an ATR Activator Involved in DNA Replication Stress Response. *Curr Biol* **26**, 3257-3268 (2016). <https://doi.org/10.1016/j.cub.2016.10.030>
- 51 Bai, L., Michael, W. M. & Yan, S. Importin beta-dependent nuclear import of TopBP1 in ATR-Chk1 checkpoint in *Xenopus* egg extracts. *Cell Signal* **26**, 857-867 (2014). <https://doi.org/10.1016/j.cellsig.2014.01.006>
- 52 Yan, S. & Willis, J. WD40-repeat protein WDR18 collaborates with TopBP1 to facilitate DNA damage checkpoint signaling. *Biochem Biophys Res Commun* **431**, 466-471 (2013). <https://doi.org/10.1016/j.bbrc.2012.12.144>
- 53 Cuadrado, M. *et al.* ATM regulates ATR chromatin loading in response to DNA double-strand breaks. *J Exp Med* **203**, 297-303 (2006). <https://doi.org/10.1084/jem.20051923>
- 54 Jazayeri, A. *et al.* ATM- and cell cycle-dependent regulation of ATR in response to DNA double-strand breaks. *Nat Cell Biol* **8**, 37-45 (2006). <https://doi.org/10.1038/ncb1337>
- 55 Stiff, T. *et al.* ATR-dependent phosphorylation and activation of ATM in response to UV treatment or replication fork stalling. *Embo J* **25**, 5775-5782 (2006). <https://doi.org/10.1038/sj.emboj.7601446>
- 56 Yan, S., Sorrell, M. & Berman, Z. Functional interplay between ATM/ATR-mediated DNA damage response and DNA repair pathways in oxidative stress. *Cell Mol Life Sci* **71**, 3951-3967 (2014). <https://doi.org/10.1007/s00018-014-1666-4>
- 57 Tell, G., Quadrioglio, F., Tiribelli, C. & Kelley, M. R. The many functions of APE1/Ref-1: not only a DNA repair enzyme. *Antioxid Redox Signal* **11**, 601-620 (2009). <https://doi.org/10.1089/ars.2008.2194>
- 58 Whitaker, A. M. & Freudenthal, B. D. APE1: A skilled nucleic acid surgeon. *DNA Repair (Amst)* **71**, 93-100 (2018). <https://doi.org/10.1016/j.dnarep.2018.08.012>
- 59 Oliveira, T. T. *et al.* APE1/Ref-1 Role in Inflammation and Immune Response. *Frontiers in immunology* **13**, 793096 (2022). <https://doi.org/10.3389/fimmu.2022.793096>
- 60 Guikema, J. E. *et al.* APE1- and APE2-dependent DNA breaks in immunoglobulin class switch recombination. *J Exp Med* **204**, 3017-3026 (2007). <https://doi.org/10.1084/jem.20071289>

REVIEWER COMMENTS

Reviewer #1 (Remarks to the Author):

Comments to the authors:

I feel they authors did a very good job of addressing my concerns through the addition of many new experiments. I feel the paper is now suitable for publication.

Comment to Reviewer #2:

The Reviewer stated "I fear that the findings are largely driven by artifacts of the experimental system. If I am mistaken, I would gladly like to read a response from the reviewers." I would like to assure the reviewer that the HSS system is a sound and reliable system to study DNA-based signaling. There are many nuances to the system that can be tricky to interpret. For example, while HSS cannot promote chromosomal duplication, as it lacks membranes and thus cannot form nuclei, it has robust "replicative" or "DNA synthesis" capacity as evidenced by its well documented ability to quantitatively convert M13 ssDNA to dsDNA. HSS can also perform all three branches of the DSB repair pathways (HR, NHEJ, and MMEJ). Lastly, the reviewer was concerned about the experiments showing ectopic activation of ATM by high concentration of APE1. It's worth noting that this is a common feature of ATM/ATR signaling as others have shown ectopic activation of ATR by high concentrations of TOPBP1 in a DNA-free manner in both frog egg extracts and intact mammalian cells. One idea is that, in vivo, ATM/ATR activators oligomerize at sites of damage to activate the kinases. Thus there is a local high concentration of the activator on DNA. To mimic this off DNA one has to raise the concentration of the activator globally to see the same effect.

Reviewer #2 (Remarks to the Author):

The authors have attempted to address my concerns. The issues to the ssDNA breaks and the use of the Xenopus system are still not very convincing, but given the limitations, I am satisfied with the answers.

However, the authors do not address my concerns of the phase separation model. Attached

with this review is a mass photometry analysis of APE1. We did this as courtesy to the authors since they do not have access to instrumentation. APE1 forms a dimer in solution and if you take out reducing agents, it forms a trimer. Interestingly, in the presence of DNA, the complex shifts to a monomer as seen in the structural studies (please see work from the Freudenthal group).

Thus, in vitro, the reaction goes the opposite trend when bound to DNA. Thus, the phase separation model proposed does not agree. I unfortunately do not agree with the phase-separation model proposed. Thus, I cannot support the paper in its current form.

Mass Photometry Reaction Conditions:

100 nM APE1 protein (no tags)

50 mM Tris acetate pH 7.5

50 mM KCl

+/- 1 mM TCEP

5 mM MgCl₂

APE1 Molecular Weight: 35.6 kDa

APE1 is a dimer in solution in the presence of TCEP and a trimer in the absence of reducing agents.

Point by point response to reviewers:

REVIEWER COMMENTS

Reviewer #1 (Remarks to the Author):

Comments to the authors:

I feel they authors did a very good job of addressing my concerns through the addition of many new experiments. I feel the paper is now suitable for publication.

Comment to Reviewer #2:

The Reviewer stated "I fear that the findings are largely driven by artifacts of the experimental system. If I am mistaken, I would gladly like to read a response from the reviewers." I would like to assure the reviewer that the HSS system is a sound and reliable system to study DNA-based signaling. There are many nuances to the system that can be tricky to interpret. For example, while HSS cannot promote chromosomal duplication, as it lacks membranes and thus cannot form nuclei, it has robust "replicative" or "DNA synthesis" capacity as evidenced by its well documented ability to quantitatively convert M13 ssDNA to dsDNA. HSS can also perform all three branches of the DSB repair pathways (HR, NHEJ, and MMEJ). Lastly, the reviewer was concerned about the experiments showing ectopic activation of ATM by high concentration of APE1. It's worth noting that this is a common feature of ATM/ATR signaling as others have shown ectopic activation of ATR by high concentrations of TOPBP1 in a DNA-free manner in both frog egg extracts and intact mammalian cells. One idea is that, in vivo, ATM/ATR activators oligomerize at sites of damage to activate the kinases. Thus there is a local high concentration of the activator on DNA. To mimic this off DNA one has to raise the concentration of the activator globally to see the same effect.

Response:

We thank Reviewer #1 for more clarifications on the unique features and characteristics about the *Xenopus* HSS system. We appreciated the proposed idea of local high concentration of APE1 at SSB sites via oligomerization in order to promote the ATM signaling. Thus, we changed the wording of "phase separation" into "oligomerization" in relevant places in our second revision.

Reviewer #2 (Remarks to the Author):

The authors have attempted to address my concerns. The issues to the ssDNA breaks and the use of the *Xenopus* system are still not very convincing, but given the limitations, I am satisfied with the answers.

However, the authors do not address my concerns of the phase separation model. Attached with this review is a mass photometry analysis of APE1. We did this as courtesy to the authors since they do not have access to instrumentation. APE1 forms a dimer in solution and if you take out reducing agents, it forms a trimer. Interestingly, in the presence of DNA, the complex shifts to a monomer as seen in the structural studies (please see work from the Freudenthal group).

Thus, in vitro, the reaction goes the opposite trend when bound to DNA. Thus, the phase separation model proposed does not agree. I unfortunately do not agree with the phase-separation model proposed. Thus, I cannot support the paper in its current form.

Response:

We thank the satisfaction of Reviewer #2 on the SSB-induced ATM signaling and the use of *Xenopus* system for this study.

We appreciated the courtesy by Reviewer #2 sharing the Mass Photometry data of untagged APE1 recombinant protein in solution. This interesting data show that APE1 protein forms dimer or trimer in solution with or without reducing agents, consistent with our APE1 oligomerization data shown in Fig. 5A-5E. It is also noted that 100 nM APE1 protein was utilized in the Mass Photometry assay. Furthermore, our data from Fig. S3G has estimated that the endogenous APE1 protein in the HSS system was ~1.5 μ M. Previous studies have indicated that endogenous *Xenopus* APE1 protein is about 1.5 μ M in *Xenopus* eggs, and that endogenous human APE1 protein is about 2.8 μ M in human kidney cell HEK293T (Wuhr et al., Curr Biol, 2014, PMID 24954049; Wisniewski et al., Mol Cell Proteomics, 2014, PMID 25225357). It is interesting in future studies to examine APE1 oligomerization status using Mass Photometry assays when APE1 protein concentration is

[figure redacted]

Additional Figure 1 for Reviewer's review only (not for publication).
PDB database for all 63 human APE1 structures with or without ligands.

increased to the range of physiological concentrations (e.g., ~1.5 - 3 μ M). Reviewer #2 raised a concern that APE1 forms monomer when combined with or in the presence of DNA in structural studies from labs such as Bret Freudenthal group. It is important to note that series of APE1 structural studies from Bret Freudenthal group have always utilized the N-terminal domain (NTD)-deletion mutant APE1 (i.e., first 43 amino acids are lacking), which is deficient for APE1 dimerization or oligomerization (shown in our **Fig. 5A**). This NTD-deletion mutant APE1 makes it impossible to illustrate the potential oligomerization of APE1 in the presence of double-strand DNA (dsDNA) containing AP site or SSB with distinct 3'-termini (e.g., mismatches or 8'-oxoG) (Freudenthal et al., *Nature Structural & Molecular Biology*, 2015, PMID 26458045; Whitaker et al., *Nature Communications*, 2018, PMID 29374164; Whitaker et al., *Nucleic Acids Res*, 2022, PMID 36018803). In addition, earlier structural characterization of APE1 in complex with dsDNA containing AP site from John Tainer group could not illustrate the structure of APE1 NTD, although the full-length APE1 recombinant protein was used for the structural study (Mol et al., *Nature*, 2000, PMID 10667800). It is worth of pointing out a long-outstanding question in the field that no structure on APE1 NTD has ever been characterized or visualized to a satisfactory or publication-quality level. As shown in **Additional Figure 1** for Reviewer's review only, we found that out of the 63 searchable/reportable human APE1 (P27695) structures with or without ligands (DNA or others) from PDB database, only AlphaFold analysis predicts that APE1 NTD extends outside of the EEP core (already shown in our **Fig. 5B**). Although APE1's EEP domain is sufficient for binding and interacting with SSB site, the findings from structural studies could not exclude the possibility that the NTD domain of full-length APE1 mediates interaction with other heterogenous APE1 molecules while its EEP domain associates with SSB site. Overall, our findings and proposed model don't disagree with previous structural studies. Nevertheless, we incorporated findings from structural studies with appropriate references, and revised our model I of ATM signaling activation by APE1 at SSB site in *Xenopus* egg extracts (**Revised Fig. 5I**).

We understand Reviewer #2 was not in favor of our proposed phase separation model. Thus, we revised our Fig. 5I and proposed APE1 oligomerization as a working model to interpret our data reported in this research project. We also changed the wording of "phase separation" to "oligomerization" or "oligomers" in relevant places in the revision.

Accordingly, we revised the two modes of ATM signaling activation by APE1 in our revised manuscript:

Mode I in Line 523-531 on Page 17:

"Therefore, our working model of how ATM is activated at SSB sites as follows (Mode I, Fig. 5I): **(Step 1)** APE1's EEP domain can sense and associate with SSB site¹⁻⁴; **(Step 2)** APE1's NTD domain may mediate APE1 oligomerization, whereas APE1 can initiate the SSB end resection in the 3'-5' direction via its EEP domain's exonuclease activity to generate a small ssDNA gap⁵; **(Step 3)** APE1's NTD domain associates with ATM and the MRN complex; and **(Step 4)** ATM kinase is recruited and activated by APE1 protein directly. Whether the conformation and/or biophysical property of APE1 monomer or oligomers at SSB site or converted gap structure are changed or transitioned warrants future studies for further clarifications."

Mode II in Line 543-551 on Page 18:

"These data support a distinct mechanism of direct activation of ATM by overexpressed APE1 in the absence of DNA in *Xenopus* egg extract (Mode II, Fig. 5I): **(Step 1)** When overexpressed, high concentration of APE1 protein can form different species of APE1 dimers and/or oligomers in *Xenopus* egg extract, which is dependent on its NTD motif; **(Step 2)** Oligomerized APE1 can recruit ATM protein into APE1 oligomers with high local concentrations in the absence of DNA in *Xenopus* egg extract; **(Step 3)** ATM kinase is directly activated by APE1 protein. Future experiments are warranted to better characterize the distinct features of APE1 oligomerization formation and disassembly by biophysical and molecular/cell biology approaches such as mass photometry technology, super-resolution microscopy analysis, and single-molecular technologies."

References:

- 1 Whitaker, A. M., Flynn, T. S. & Freudenthal, B. D. Molecular snapshots of APE1 proofreading mismatches and removing DNA damage. *Nat Commun* **9**, 399 (2018). <https://doi.org/10.1038/s41467-017-02175-y>
- 2 Whitaker, A. M., Stark, W. J. & Freudenthal, B. D. Processing oxidatively damaged bases at DNA strand breaks by APE1. *Nucleic Acids Res* **50**, 9521-9533 (2022). <https://doi.org/10.1093/nar/gkac695>
- 3 Freudenthal, B. D., Beard, W. A., Cuneo, M. J., Dyrkheeva, N. S. & Wilson, S. H. Capturing snapshots of APE1 processing DNA damage. *Nat Struct Mol Biol* **22**, 924-931 (2015). <https://doi.org/10.1038/nsmb.3105>
- 4 Mol, C. D., Izumi, T., Mitra, S. & Tainer, J. A. DNA-bound structures and mutants reveal abasic DNA binding by APE1 and DNA repair coordination [corrected]. *Nature* **403**, 451-456 (2000). <https://doi.org/10.1038/35000249>
- 5 Lin, Y. *et al.* APE1 senses DNA single-strand breaks for repair and signaling. *Nucleic Acids Res* **48**, 1925-1940 (2020). <https://doi.org/10.1093/nar/gkz1175>